



# Estimating Upper Silesian coal mine methane emissions from airborne in situ observations and dispersion modeling

Julian Kostinek[1], Anke Roiger[1], Maximilian Eckl[1], Alina Fiehn[1], Andreas Luther[1], Norman Wildmann[1], Theresa Klausner[1], Andreas Fix[1], Christoph Knote[2], Andreas Stohl[3], and André Butz[4]

[1]Deutsches Zentrum für Luft- und Raumfahrt (DLR), Institut für Physik der Atmosphäre, Oberpfaffenhofen, Germany
[2]Meteorological Institute, Ludwig-Maximilians-University Munich, Munich, Germany
[3]Department of Meteorology and Geophysics, University of Vienna, Vienna, Austria
[4]Institute of Environmental Physics, University of Heidelberg, Heidelberg, Germany

**Correspondence:** Julian Kostinek (julian.kostinek@dlr.de)

**Abstract.** Abundant mining and industrial activities located in the Upper Silesian Coal Basin (USCB) lead to large emissions of the potent greenhouse gas (GHG) methane ($CH_4$). The strong localization of $CH_4$ emitters (mostly confined to known coal mine ventilation shafts) and the large emissions of 448 / 720 kt $CH_4$ yr$^{-1}$ reported in the European Pollutant Release and Transfer Register (E-PRTR 2017) and the Emissions Database for Global Atmospheric Research (EDGAR v4.3.2) make the USCB a prime research target for validating and improving $CH_4$ flux estimation techniques. High-precision observations of this GHG were made downwind of local (e.g. single facilities) to regional-scale sources (e.g. agglomerations) in the context of the CoMet 1.0 campaign in early summer 2018. A Quantum Cascade/Interband Cascade Laser (QCL/ICL) based spectrometer adapted for airborne research was deployed aboard the German Aerospace Centers (DLR) Cessna 208B to sample the planetary boundary layer (PBL) in situ. Regional $CH_4$ emission estimates for the USCB are derived using a model approach including assimilated wind soundings from three ground-based Doppler lidars. Although retrieving estimates for individual emitters is difficult using only single flights, due to sparse data availability, the combination of two flights allows for exploiting different meteorological conditions (analogous to a sparse tomography algorithm) to establish confidence on facility level estimates. Emission rates from individual sources are not only needed for unambiguous comparisons between bottom-up and top-down inventories but become indispensable if (independently verifiable) sanctions are to be imposed on individual companies emitting GHGs. An uncertainty analysis is presented for both the regional scale and facility level emission estimates. We find instantaneous emission estimates of 452 / 442 ± 78 / 75 kt $CH_4$ yr$^{-1}$ for the morning / afternoon flight of June 6th, 2018. The derived emission rates coincide (±2 %) with annual-average inventorial data from E-PRTR 2017 albeit they are distinctly lower (-37 % / -40 %) than values reported in EDGAR v4.3.2. Discrepancies in available emission inventories could potentially be narrowed down with sufficient observations using the method described herein to bridge the gap between instantaneous emission estimates and yearly averaged inventories.





## 1 Introduction

The growth in population and economy since the pre-industrial era has been going hand in hand with rising anthropogenic emissions, causing a strong increase in atmospheric greenhouse gas (GHG) concentrations. This is in general attributed to anthropogenic emissions from a large variety of sources omnipresent in modern life (Pachauri et al., 2014), e.g. extraction, processing and transport of fossil fuels. Despite the evident anthropogenic influence on the climate in general, large uncertainties remain in the magnitude of human induced radiative forcing and relative contributions from different sectors (Nisbet et al. (2014); Kirschke et al. (2013)). Approximately $20\%$ of the global $CH_4$ source are estimated to arise from the sector of fossil fuel industry (Schwietzke et al., 2016), that also includes activities like coal mining - an industry for which the Upper Silesian Coal Basin (USCB), located in Southern Poland and the Czech Republic, is well known for.

According to the European Pollutant Release and Transfer Register (E-PRTR 2017, https://prtr.eea.europa.eu/) a total of $448 \, \mathrm{kt} \, CH_4 \, \mathrm{yr}^{-1}$ is emitted into the air from the USCB region, making it one of Europe's methane emission hot spots. The intensive mining activities and the heavy industry spread around the city of Katowice lead to these significant amounts of $CH_4$ emitted into the atmosphere, where over $99\%$ of the $CH_4$ emissions reported in E-PRTR 2017, listing emitters above a threshold of $0.1 \, \mathrm{kt} \, CH_4 \, \mathrm{yr}^{-1}$, are attributed to mining and related industry. These large emissions are also apparent in the EDGAR v4.3.2 emission inventory reporting a total of $720 \, \mathrm{kt} \, CH_4 \, \mathrm{yr}^{-1}$ in 2012.

The design and subsequent control of mitigation measures to slow down the increase in atmospheric GHG concentrations requires reliable verification and attribution of GHG emissions now and in the future. An established method to derive GHG emissions is known as the top-down approach. This method is based on observed GHG concentrations in the atmosphere and projects their variations (both in time and space) back onto the emissions that may have caused these variations (Nisbet and Weiss (2010); Chevallier et al. (2005); Peters et al. (2007)). For $CH_4$, concentrations downwind of local- to regional-scale emission sources can be sampled efficiently using high-precision airborne measurements within the planetary boundary layer (PBL). On-board meteorological instrumentation allows for concurrent sensing of important atmospheric state variables like static pressure and air temperature, as well as the local wind field, which are particularly useful to estimate emissions. In situ instruments provide point measurements at high precision that can well be used for flux estimation using techniques like the mass balance approach (Karion et al.; Conley et al. (2017); Pitt et al. (2018)). Previous studies used airborne in situ instrumentation to estimate regional $CH_4$ emissions from oil and natural gas operations in the U.S. and Canada (Johnson et al. (2017); Karion et al. (2015); Barkley et al. (2017)). These studies find emission inventories (EDGAR) to underestimate $CH_4$ emissions from the respective sector. Recent studies also targeted urban $CH_4$ emissions (Ryoo et al. (2019); Plant et al. (2019); Ren et al. (2018)) and anthropogenic $CH_4$ emissions from agriculture and waste treatment (Yu et al., 2020). Airborne in situ data have further been used to estimate emissions on facility level by flying closed circles around individual emitters (Lavoie et al. (2015); Conley et al. (2017); Hajny et al. (2019); Mehrotra et al. (2017); Baray et al. (2018)).

Recently, Luther et al. (2019) reported on XCH4 flux estimates ranging from $6 \pm 1 \, \mathrm{kt} \, CH_4 \, \mathrm{yr}^{-1}$ for single shafts and up to $109 \pm 33 \, \mathrm{kt} \, CH_4 \, \mathrm{yr}^{-1}$ for a subregion of the USCB from ground-based, portable, sun-viewing Fourier transform spectrometers mounted on a truck. Fiehn et al. (2020) investigated $CH_4$ emissions from the USCB using a mass balance approach. They report





emission estimates of $436 \pm 115\,\text{kt CH}_4\,\text{yr}^{-1}$ and $477 \pm 101\,\text{kt CH}_4\,\text{yr}^{-1}$ from two research flights along with a detailed uncertainty analysis. The present study is based on the very same research flights and aims at contributing an advanced model approach. Previous studies have used Lagrangian models to simulate the dispersion (Tuccella et al. (2017); Raut et al. (2017)) of plumes emanating from oil and gas platforms or identification of $CH_4$ sources (Platt et al., 2018). Atmospheric transport models have been used to infer $CH_4$ emissions from the oil and natural gas industry (Barkley et al., 2017). Here, a combination

of a Eulerian atmospheric transport model and a Lagrangian particle dispersion model is used in conjunction with assimilated Doppler lidar soundings to infer instantaneous $CH_4$ emissions for Europe's largest coal extraction region, the USCB.

Section 2 provides an introduction on the USCB as the region of prime interest followed by a research flight overview in Sect. 3. Section 4 details a model based flux estimation approach. $CH_4$ emission estimates will be given in the form of a case study in Sect. 4.1 for two research flights on June 6th, 2018 along with an estimate of the uncertainties involved. Section 5

summarizes our findings and concludes the study.

## 2  The Upper Silesian Coal Basin

The Upper Silesian Coal Basin is a plateau elevated between $200\,\text{m}$ and $300\,\text{m}$ above sea level in southern Poland. To the south it is confined by the Tatra Mountains reaching up to $2655\,\text{m}$ above sea level and forming a natural border between Slovakia and Poland. To the west it extends across the national border between Poland and the Czech Republic into the Ostrava region.

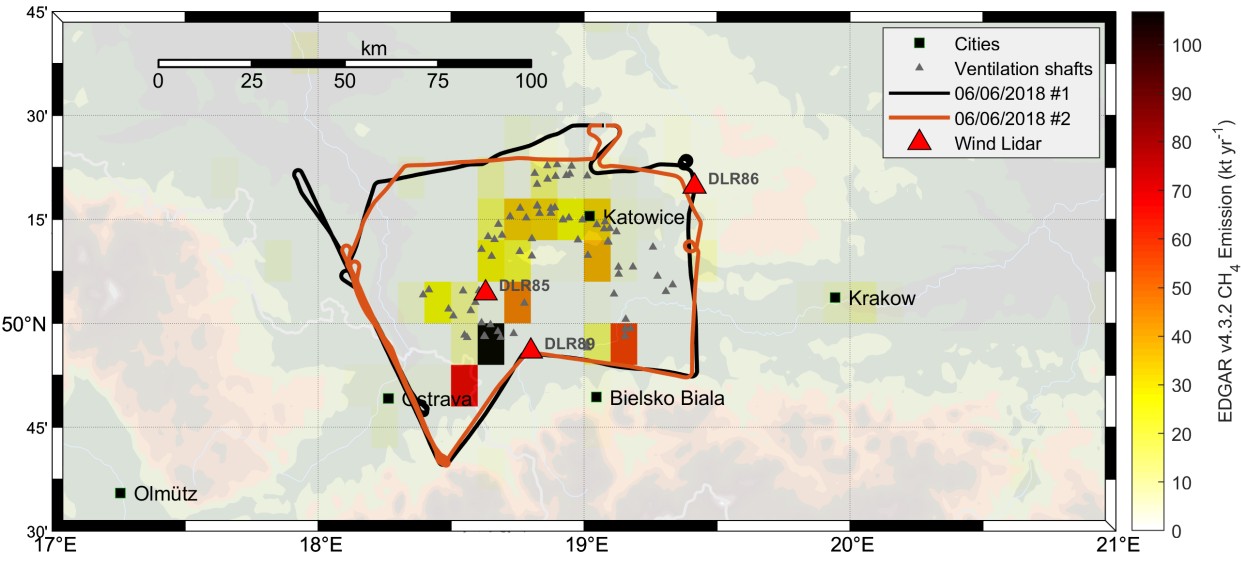

**Figure 1.** Flight trajectories of two flights of the DLR Cessna 208B sampling the USCB area during the CoMet field campaign. The plot shows flight trajectories for the morning flight (black) and afternoon flight (red) on June 6th, 2018. Red triangles mark the location of three Doppler wind lidars, deployed in the USCB area during the CoMet campaign. Grey triangles mark known coal mine ventilation shafts. Colored tiles are from the EDGAR v4.3.2 $CH_4$ emission inventory for 2012 showing typical emissions ranging up to $\sim100\,\text{kt yr}^{-1}$.





According to Gzyl et al. (2017), the USCB is well known for its abundant mining and industrial activities, including coal, zinc and lead ore exploitation. Coal mining activities make up for the largest part, with an approximate total of 10 billion metric tonnes extracted since the industrial revolution, where over 70 % of this exploitation took place after 1945. To date an approximate 75 million tonnes of coal are extracted every year from 27 active mines. It is these figures and the large area of approximately $7400\,\mathrm{km}^2$ covered, that make the USCB the largest coal extraction region in Europe (Dulias, 2016). The

intensive coal mining activities and the heavy industry spread around the city of Katowice, Poland, located in the north of the USCB, lead to significant amounts of GHG emissions in this area. Fugitive $CH_4$ emanating from the coal mine shafts reaching several hundred meters into the ground is either actively ventilated (active mines) or degasses passively from abandoned mines. Mines located in the north of the USCB are mostly abandoned and partially flooded, while intensive, active coal exploitation is located in the southern USCB, both in Poland and the Czech Republic (Gzyl et al., 2017).

Global emission inventories show large sources of methane in this area as depicted by the colored tiles in Fig. 1. The Figure is based on a subset of the publicly available EDGAR v4.3.2 $CH_4$ (https://edgar.jrc.ec.europa.eu/) emission inventory (Janssens-Maenhout et al., 2017). It shows $CH_4$ emissions range up to approx. $100\,\mathrm{kt\,yr}^{-1}$ on a $0.1 \times 0.1$ degree grid with source strengths increasing towards the southern USCB. Accordingly, the strongest sources are located near the Czech border mid ways between the cities of Bielsko-Biala, Poland and Ostrava, Czech Republic. According to EDGAR v4.3.2, these $CH_4$ sources are among

the strongest in Europe. The total $CH_4$ emissions from this inventory amount to approximately $720\,\mathrm{kt\,yr}^{-1}$ for the USCB region, where $\sim 620\,\mathrm{kt\,yr}^{-1}$ are attributed to the fuel exploitation sector. Facility level emission data of $CH_4$ are provided by E-PRTR 2017. The locations of 74 documented coal mine ventilation shafts (active and inactive) have been added to Fig. 1 for reference. These locations were visually identified from satellite imagery and emission values from E-PRTR 2017 were evenly distributed among the ventilation shafts for each company (see also Nickl et al. (2019)). According to E-PRTR 2017 individual

contributions sum up to a total $CH_4$ emission of approximately $448\,\mathrm{kt\,yr}^{-1}$. This value is approximately 38 % lower compared to the EDGAR v4.3.2 inventory (28 % if only considering the fuel exploitation sector), showing the large uncertainties present in the available data.

## 3   Research flight overview

The CoMet mission in early summer 2018 primarily aimed at providing observations of GHG (mainly $CO_2$ and $CH_4$) gradients

along large-scale latitudinal transects over Europe from co-ordinated operation of several state-of-the-art instruments on ground and aboard five research aircraft. Aboard the Cessna 208B, a rich dataset of simultaneous airborne observations of $CH_4$, $C_2H_6$, $CO_2$, CO, $N_2O$ and $H_2O$ was collected using the QCLS instrument (see Fig. 2 and Kostinek et al. (2019) for details) during $\sim$30 flight hours. In the following, a subset of these data from two research flights undertaken on June 6th, 2018 was used to retrieve $CH_4$ fluxes emanating from the USCB region. Both flight tracks are shown in Fig. 1 along with the locations of three

co-deployed Leosphere Windcube 200S Doppler wind lidars (Wildmann et al., 2020). The morning (black line) and afternoon flights (red line) circumvent all known ventilation shafts in the area (gray triangles) and are in fact very similar (congruent) from the top-down perspective. This is well intended to enhance confidence on retrieved GHG fluxes. Moderate ($3 - 6\,\mathrm{m\,s}^{-1}$)





winds throughout the day from north-easterly directions drive advection of the $CH_4$ plumes towards the Czech border and into the Ostrava region.

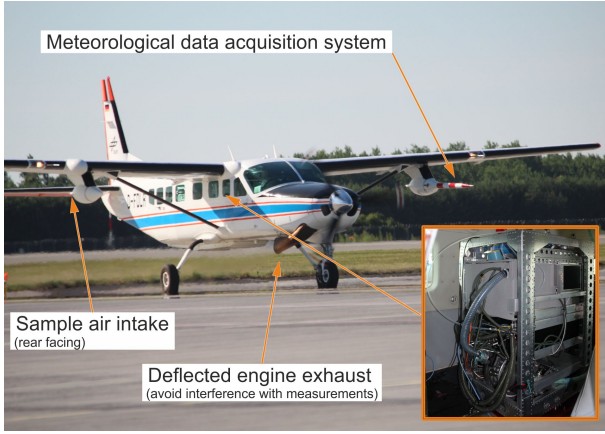

**Figure 2.** The DLR Cessna 208B on the taxiway at Katowice airport. The sample air intake is mounted underneath the right wing. The meteorological data acquisition system underneath the left wing. The QCLS instrument (lower left panel) is located inside the cabin behind the pilot seats.

The morning flight (black line in Fig. 1) on June 6th, 2018 starts off from Katowice airport, located to the north of the city center, around 0915 UTC. Following a short constant-altitude transect a spiral-up was flown in the east to get a sounding out of the boundary layer. This maneuver, revealing a boundary layer depth of approximately 1150 m above mean sea level (a.M.S.L), was followed by an upwind leg flown at a constant altitude of 900 m a.M.S.L showing a fairly homogeneous $CH_4$ inflow into the area of interest, thus allowing for subtracting from the measured mole fractions downwind of the mines. Mixing

ratios decreased slightly towards free tropospheric background values when climbing above the PBL. Before returning back to Katowice airport at around 1145 UTC, the downwind wall maneuver, consisting of 5 constant-altitude flight legs to the west was performed at altitudes of approximately 800 m, 1.1 km, 950 m, 1.4 km, and 1.8 km a.M.S.L, respectively. During the last two flight legs the aircraft was outside of the PBL. This allows for verifying if entrainment / detrainment from the PBL into the free troposphere is marginal enough to be disregarded.

The afternoon flight (red line in Fig. 1) started off from Katowice airport around 1315 UTC. An upwind leg flown at a constant altitude of 900 m a.M.S.L again showed a fairly homogeneous $CH_4$ inflow into the USCB area. Mixing ratios decrease slightly towards free tropospheric background values when climbing above the PBL during a climb and descent maneuver flown parallel to the sensed mean wind direction. Before returning back to Katowice airport at around 1530 UTC, the downwind wall maneuver, consisting of 6 constant-altitude flight legs was performed over the western USCB region at altitudes of

approximately 800 m, 890 m, 975 m, 950 m, 1.1 km, 1.5 km, and 1.8 km a.M.S.L, respectively.



## 4 Estimating emissions

The model-based approach developed in this work employs a combination of Eulerian and Lagrangian particle dispersion models. Due to the known locations of the coal mine ventilation shafts, their emissions are modeled forward in time with constant emission rates. Modeled data are then extracted at the aircraft position in space and time and compared to actual airborne in

situ observations. This comparison depends on the quality of the a-priori emission data, the quality of the measurements and the quality of the transport model simulation, which in turn depends on the quality of the meteorological data (winds, PBL heights, etc.). Validating the meteorological data is therefore important to enable regional emission estimates based on particle dispersion models. Here, meteorological driver data is generated using the Weather Research and Forecasting (WRF) v4.0 model (Powers et al., 2017) with assimilated soundings from three Leosphere Windcube 200S Doppler wind lidars. Data is then

fed into the Lagrangian particle dispersion model FLEXPART-WRF ("FLEXible PARTicle dispersion model") - a FLEXPART (Pisso et al., 2019) flavour adapted for WRF meteorology - and used to model the exhaust plumes emanating from ventilation shafts of the emitters listed in E-PRTR 2017.

### 4.1 Local scale meteorology using WRF

Figure 3 shows a satellite map of central Europe with the two domains specified for the USCB region. The outer domain D1 (light blue box in Fig. 3) with a horizontal grid resolution of $\sim 15\,\mathrm{km}$ includes large parts of central Europe. This domain is fed

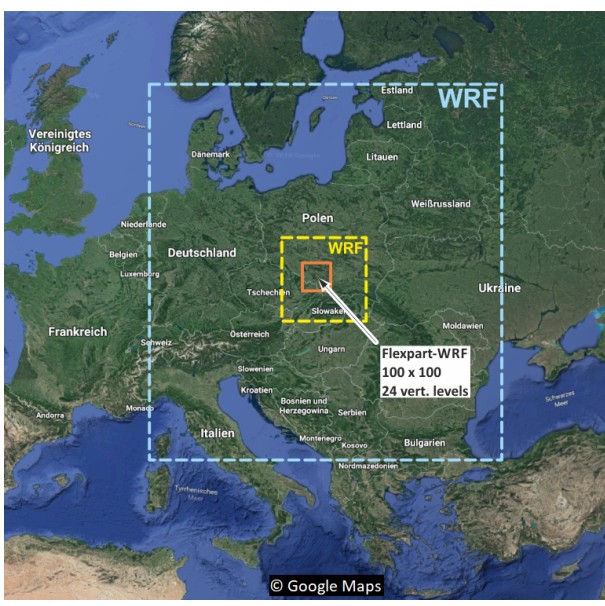

**Figure 3.** The FLEXPART-WRF domain resides in the nested WRF domain D2 providing the meteorological driver data. Generous spacing towards the driver domain has been included to avoid spurious boundary effects.


by NCEP GDAS/FNL Operational Global Analysis data on a $0.25$-degree x $0.25$-degree grid, available from the NCAR/UCAR





Research Data Archive at 3 hours time resolution (GDAS/FNL, 2015). The grid four dimensional data assimilation (GFDDA) module is used to nudge modeled meteorology towards the analysis data at each grid point. The outer domain is intended to catch the large scale weather situation over Europe and to provide a smooth transition between the coarse NCEP GDAS/FNL

Operational Global Analysis and the region of prime interest. The inner domain D2 (yellow box in Fig. 3) has a horizontal grid resolution of $\sim 3\,\mathrm{km}$ and covers the entire USCB region. The model output from D2 is the primary product required for subsequent FLEXPART runs. Both domains are driven with the original WRF v4.0 topographic data with a resolution of 30 arc-seconds. Vertically, the model atmosphere is divided into 33 stacked layers, with the top layer at 200 hPa (corresponding to approximately 12 km altitude). Vertical layers are closer spaced at lower altitudes to enable a better resolution of boundary

layer processes. The modeled atmospheric state variables are output every hour for D1 and every 5 minutes for D2.

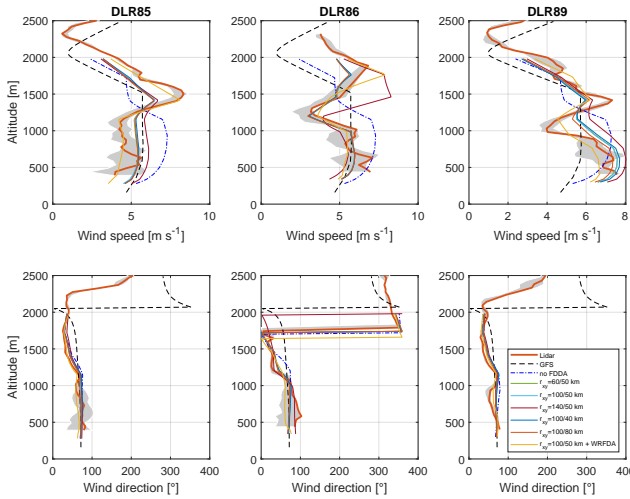

**Figure 4.** Ensemble WRF runs with varying radius of influence $r_{xy}$ in comparison to interpolated NCEP GDAS/FNL and actual Lidar soundings for June 6th, 2018 at 0900 UTC. The shaded area shows the maximum variability including soundings timed 20 minutes before and 20 minutes after the observations used. Abscissa units are ms$^{-1}$ and degrees, respectively.

Soundings from three Doppler lidars (marked DLR85, DLR86 and DLR89) deployed in the USCB area during the CoMet mission (see Fig. 1 for respective positions) have been used to augment the model output (Wildmann et al., 2020). These data are available on a regular, continuous basis throughout the campaign period at 10 min time intervals with soundings typically reaching up to $\sim$2.5 km a.M.S.L depending on the atmospheric condition. Domains D1 and D2 are both nudged towards

the Doppler soundings using the WRF-FDDA subsystem (Deng et al., 2008). Sensitivity of the model output on three key parameters of the observational data assimilation subsystem, namely the radius of influence $r_{xy}$, time window $\Delta_t$ and horizontal wind coefficient $c_{uv}$ were analyzed through numerous runs with the goal of finding an appropriate configuration. Figure 4 shows ensemble runs with varying $r_{xy}$ in comparison to interpolated NCEP GDAS/FNL and the actual lidar soundings for June 6th, 2018 at 0900 UTC. The shaded gray area beneath the orange-colored lidar soundings shows the maximum variability including

soundings timed 20 min before and 20 min after the observations used. Figure 4 demonstrates that modeled data are in good




agreement to observed Doppler soundings when using WRF-FDDA. It also shows discrepancies between NCEP GDAS/FNL driver data and observations in wind direction and more importantly on the wind speed in the lower troposphere and the PBL depth. To further enhance compatibility between model and observations, the WRFDA submodule (Barker et al., 2012) was used in 3DVar cycling mode similar to Liu et al. (2013) using the NCAR CV3 background error covariance (Barker et al.,

2004). The required observational error covariances are taken from the measurement uncertainties.

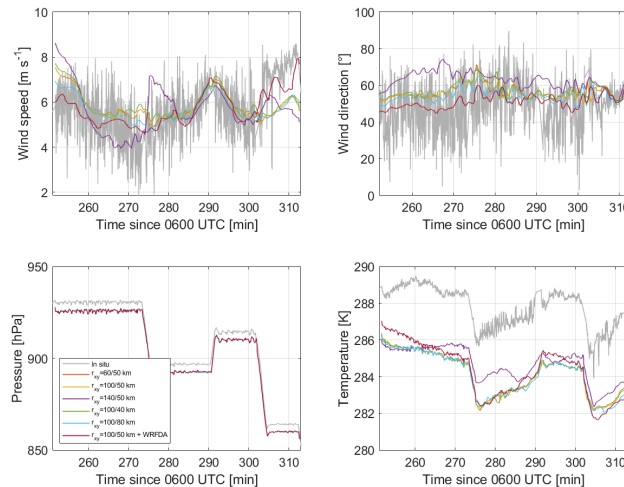

**Figure 5.** Comparison of 1 Hz wind speed (upper left), wind direction (upper right), static pressure (lower left) and static air temperature (lower right) as measured aboard the Cessna 208B on June 6th, 2018 between 1000 UTC and 1120 UTC to ensemble runs with varying $r_{xy}$ from above. The graphs are plotted as a function of time in minutes since 0600 UTC.

    To verify and validate the observational-FDDA approach, non-assimilated meteorological in situ data collected aboard the Cessna 208B are compared to modeled data in Fig. 5. In particular, Fig. 5 compares 1 Hz wind speed, wind direction, static pressure and static air temperature as measured on June 6th, 2018 between 1000 UTC and 1120 UTC with the underwing boom-mounted data acquisition system to ensemble runs with varying $r_{xy}$ from above. These data were collected approximately

35 km (minimum distance) to 65 km (maximum distance) to the west of the nearest wind lidar during the downwind wall phase of the morning flight (see Fig. 1). Modeled data, extracted at the aircraft positions in space and time, agree with observations of wind speed and direction to within $\pm\,0.7\,\mathrm{m\,s^{-1}}$ (1 $\sigma$) and $\pm\,5\,^\circ$ (1 $\sigma$), respectively. Here, the NCAR Command Language (NCL, Brown et al. (2012)) has been used to interpolate from gridded model output to the exact aircraft position in space and time. Modeled wind speed deviates from observed winds during the last 20 minutes of the downwind wall. A possible reason

for this might be that the flight leg was in close vicinity to the PBL top height. A bias of modeled static pressure and static air temperature is evident from the lower panels in Fig. 5. Modeled pressure has a consistent offset of -5 hPa compared to in situ data and modeled temperature is biased approximately 2.2 K towards lower values.


## 4.2 Plume dispersion using FLEXPART-WRF

FLEXPART-WRF version 3.3.2 (Brioude et al., 2013) was used to model the exhaust plumes of known emitters forward in
time using the meteorological data obtained from the WRF simulations described above (see Sect. 4.1) as a driver. The model
is set to release 50 000 particles and an arbitrarily chosen total mass of $m_e = 1 \times 10^5$ kg for each release location during the
total simulated time of $\tau_e = 9$ hours. The model output is gridded into $100 \times 100$ horizontal tiles and 24 vertically stacked
layers ranging from ground level up to 3 km in altitude. This results in a horizontal resolution of approximately 1.3 km and a
vertical resolution of 50 m near ground, gradually increasing to 500 m above 2 km altitude. The domain has been placed inside
the nested WRF domain D2 with generous spacing towards the domain boundaries as indicated in Fig. 3 to avoid spurious
boundary effects. The main product of the FLEXPART-WRF runs are concentration fields for each release location in units of
ng m$^{-3}$, which are scaled a-posteriori to deduce the emission rate of each modeled release. Each coal mine ventilation shaft
is modeled as a constant, continuous volume source $\varphi_i$ with a 10 m $\times$ 10 m horizontal footprint and extending 10 m in the
vertical direction. The volume emitter sizes are based on the construction of typical ventilation shafts in the USCB (Swolkień,
185 2020).

Mass densities in units of ng m$^{-3}$ can be extracted for the aircraft positions from the model output. The result is a $m \times n$
linear forward model matrix $K_{ji}$ that links scaling factors for emission rates to atmospheric mass density enhancements at the
measurement instances, where $m$ is the number of observations available and $n$ is the number of modeled release locations, i.e.
$K_{ji}$ is the mass density that source $i$ contributes to observation $j$. A scaling coefficient $x_i$ is assigned to each of the $n$ sources
$\varphi_i = m_e \tau_e^{-1}$, with the total emission time $\tau_e$ in seconds and the total mass emitted $m_e$ in kg. These last two parameters are
both assigned in the FLEXPART-WRF input file. Following a least-squares approach, the scaling coefficients $x_i$ can be found
through minimization of the squared difference between observations and model output for each of the $n$ modeled sources $\varphi_i$
and for each of the $m$ observed enhancements $y_j$ available

$$\min_x \sum_{j=1}^m \left( y_j - \sum_{i=1}^n K_{ji} x_i \right)^2$$
$$= \min_x (\mathbf{y} - \mathbf{Kx})^T (\mathbf{y} - \mathbf{Kx}) \tag{1}$$

The total emission estimate $\Phi$ in units kg s$^{-1}$ follows from the scaled sum of the individual contributions $\varphi_i$

$$\Phi = \sum_{i=1}^n x_i \varphi_i \tag{2}$$

For use cases where no a-priori information on emitters is available, the above equations directly yield the best total emission
estimate $\Phi$. In the present case however, a-priori information on the emissions of the individual shafts can be used to compute
a maximum a posteriori (MAP) solution $\mathbf{x}$ based on the observation vector $\mathbf{y}$ and the a-priori vector $\mathbf{x_a}$ (Rodgers, 2000).
Following Bayes' theorem the MAP solution is given by the minimum of the cost function (Tarantola (2004); Jacob (2007))

$$J(\mathbf{x}) = (\mathbf{x} - \mathbf{x_a})^T \mathbf{S_a}^{-1} (\mathbf{x} - \mathbf{x_a})$$
$$+ (\mathbf{y} - \mathbf{Kx})^T \mathbf{S_\epsilon}^{-1} (\mathbf{y} - \mathbf{Kx}) \tag{3}$$





with later defined a-priori and observational error covariance matrices $\mathbf{S_a}$ and $\mathbf{S_\epsilon}$, respectively. The MAP solution can be found by solving for $\nabla_x J(\mathbf{x}) = 0$ and is given by

$$\hat{\mathbf{x}} = \mathbf{x_a} + \mathbf{G}(\mathbf{y} - \mathbf{Kx}) \tag{4}$$

with the gain matrix

$$\mathbf{G} = \mathbf{S_a K^T}\left(\mathbf{K S_a K^T} + \mathbf{S_\epsilon}\right)^{-1} \tag{5}$$

By exploiting the averaging kernel $\mathbf{A} = \mathbf{GK}$ the number of degrees of freedom for signal $d_s$ can be computed as

$$d_s = tr(\mathbf{A}) \tag{6}$$

This number describes the reduction in the normalized error on $\mathbf{x}$ introduced by the available observations and hence provides a measure for the improvement in knowledge of $\mathbf{x}$, relative to the a-priori, due to the observations.

Here, the Non-Negative Least Squares (NNLS) algorithm (Lawson and Hanson, 1995) has been used to minimize the MAP cost function subject to the constraint $\mathbf{x} > 0$. This constraint is equivalent to the absence of negative sources. The NNLS algorithm solves the constrained least squares problem by splitting into active and passive subsets, where active and passive refer to the state of the constraint. The algorithm subsequently solves the unconstrained least squares problem for the passive set.

### 4.3 Estimating total uncertainty

The outlined approach is based on assumptions, of which the most important ones are: a constant emission rate over the timescale of transport from the source to the aircraft, an appropriate atmospheric background vector $\mathbf{b}$ (used to compute $CH_4$ enhancements $\mathbf{y} = \rho - \mathbf{b}$ from measured mass densities $\rho$), long lifetime of the species of interest, i.e. no chemical and physical removal on the timescale of a flight and the model being able to adequately represent the meteorological state variables. To assess uncertainty on the retrieved emission rates, several variables have been selected as most influencing systematic error sources: wind speed, wind direction, PBL height, source dislocation and an error in sensed mole fractions, that is further intended to include an error due to chosen background. Individual contributions of these error sources to total uncertainty can be identified from ensemble model runs with systematically perturbed parameters.

In addition to derived systematic uncertainties, statistical errors related to the MAP fit are to be acknowledged. The statistical uncertainty $\epsilon_i$ in the retrieved parameters $x_i$ can be expressed in terms of the parameter covariance matrix $\hat{\mathbf{S}}$ as

$$\epsilon_i = \sqrt{\hat{\mathbf{S}}_{ii}} \tag{7}$$

The parameter covariance matrix $\hat{\mathbf{S}}$ is computed from the $m$-by-$n$ dimensional forward model $\mathbf{K}$ using

$$\hat{\mathbf{S}} = \left(\mathbf{K}^T \mathbf{S_\epsilon}^{-1} \mathbf{K} + \mathbf{S_a}^{-1}\right)^{-1} \tag{8}$$





The observational error covariance matrix $\mathbf{S}_\epsilon$ has been estimated as a diagonal matrix with the squared observation uncertainties $\sigma^2$ on its main diagonal inflated with a transport model error $\chi_m$ obtained from ensemble runs with perturbed parameters (see Sect. 4.6).

$$\mathbf{S}_\epsilon = diag\left(\sigma^2 + \chi_m^2\right) \tag{9}$$

The measurement uncertainties $\sigma_i$ were obtained via standard error propagation from the uncertainties associated with different instruments aboard the aircraft needed for the computation of the $CH_4$ mass density observations (see Eq. 10). Static air temperature can be probed with an uncertainty of $\sigma_T = 0.15\,K$, static air pressure with $\sigma_p = 1\,hPa$ and wind speed with $\sigma_u = 0.3\,ms^{-1}$ (1s-1$\sigma$, Mallaun et al. (2015)). $CH_4$ mole fractions were sampled with a total uncertainty better than $1.85\,ppb$ (1s-1$\sigma$). The a-priori error covariance matrix $\mathbf{S}_\mathbf{a}$ is a diagonal matrix with the squared a-priori uncertainties, estimated with 50 % of the nominal value on its main diagonal. Finally, the Jacobian $\nabla_x J(\mathbf{x})$ is the first derivative of the cost function with respect to the parameters $x_i$. As such it describes the change in residuals introduced by a change in parameter $x_i$.

### 4.4   Case study: June 6th, 2018

Figure 6 shows a time series of the measured and modeled $CH_4$ mass density enhancement as a function of flight time during the downwind wall phase with the atmospheric background subtracted and source coefficients $x_i$ already optimized (according to Eq. 1). The measured scalar mass density $\rho$ has been deduced from the ideal gas law $pV = mR_sT$ (mass m, specific gas constant

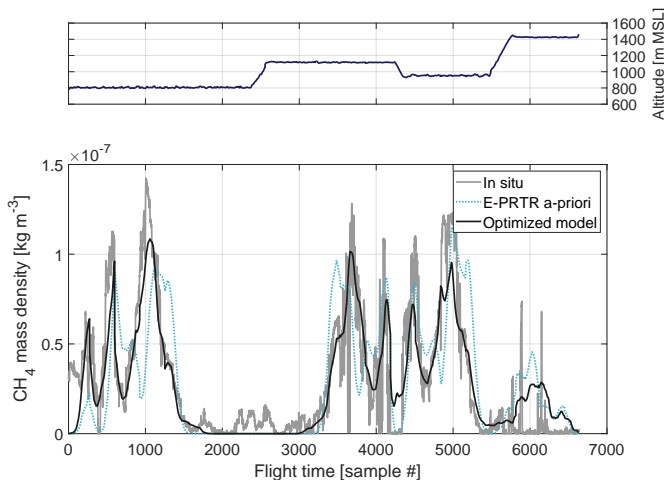

**Figure 6.** Time series of the in situ measured (gray) and modeled $CH_4$ mass density (solid black) with subtracted background as a function of flight time during the downwind wall phase of the morning flight on June 6th, 2018 with optimized source coefficients $x_i$. The dotted light blue line corresponds to the same forward simulation using scaling coefficients deduced from E-PRTR 2017.

$R_s = R/M$ and molar mass of $CH_4$ M) using the in situ measured static air temperature and static air pressure according to

$$\rho_x = \frac{m_x}{V_{air}} = \frac{m_x}{m_{air}}\left(\frac{p}{RT}\right)_{air} \tag{10}$$





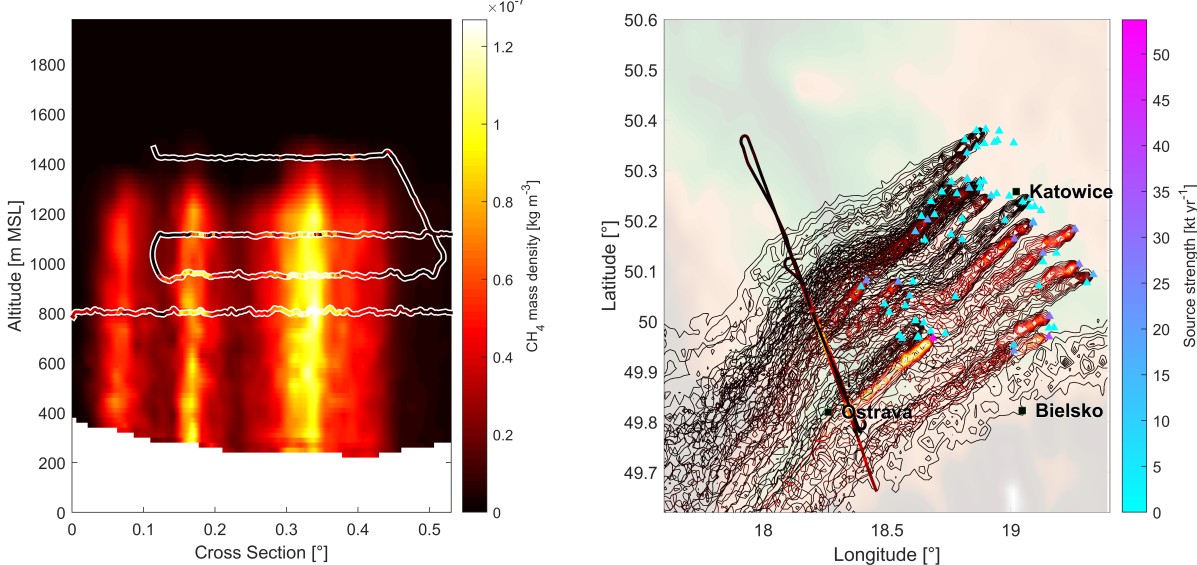

**Figure 7.** Left: Interpolated cross section of the model output along the downwind wall including scattered in situ observations of $CH_4$ for the morning flight on June 6th, 2018. The colorbar also applies to the isolines on the right. Right: Top-down view on the model output and the downwind wall observations $\rho$ at 750 m a.g.l. Triangles mark simulated emitters with colors corresponding to the optimized source strengths. Both panels show a snapshot of the model output at one fixed time chosen as the center time of the downwind wall phase.

where $m_x$ denotes the total mass of the species of interest. The unit-less coefficient $m_x \, m_{air}^{-1} = c_x \, M_x \, M_{air}^{-1}$ is obtained from the sensed $CH_4$ mole fractions $c_x$ in units mol mol$^{-1}$.

Prior to conversion, the atmospheric background has to be subtracted from the observed $c_x$. The choice of background is to some extent ambiguous, because there is no clear edge between background and in-plume sampling. This contributes to total flux estimation uncertainty, as will be discussed later. Here, a piecewise linear interpolation between the outermost boundaries of each of the 4 flight legs (see Fig. 7) has been considered as the best guess of atmospheric background. The mean value of 20 samples has been used on both edges of each flight leg. Using this approach, latitudinal and longitudinal gradients in background $CH_4$ mole fractions are accounted for by using both edges of each flight leg. Vertical gradients in background $CH_4$ levels are accounted for by treating each constant-altitude flight leg separately.

From Fig. 6 a good overall match between model (solid black line) and in situ observations (gray) is apparent with a mean bias of $2.5 \times 10^{-9}$ kg m$^{-3}$ and a root mean square error of $1.6 \times 10^{-8}$ kg m$^{-3}$. Some of the minor structure is not reproduced in detail by the model, which is expected due to the model's 3 km horizontal grid resolution. The reason for the discrepancy between model and the first few hundred observations in Fig. 6 becomes more obvious when looking at the 2D scene shown in Fig. 7. The left panel of Fig. 7 shows a cross section of the model output along the downwind wall including the in situ observations of $\rho$. The right panel of Fig. 7 depicts the top-down view on the model output and the downwind wall observations





at a fixed altitude of $750\,\mathrm{m}$ a.g.l. It should be noted here that both panels show a snapshot of the model output at one fixed time chosen as the center time of the downwind wall.

The discrepancy between model and observation at the lowermost (first in time) flight leg, corresponding to the southernmost trajectory section in Fig. 7 (right panel) can not be reproduced by any of the included emission sources. A possible source is urban $CH_4$ emissions of Krakow, located to the east of the USCB region. An area source, covering the greater city area, has therefore been included in the model. Its influence can be seen at the rightmost edge of Fig. 7 (right panel). Although we identified Krakow city as a possible source, we omitted it in the USCB emission estimates, as it does not officially belong to the USCB area. There are other parts in the time series, where the model does either underestimate (e.g. times around observation

numbers 2000-3000, 4000-4500) or overestimate emissions (e.g. around observation number 6000). This might well be related to sources not taken into account or deficiencies in wind speed, wind direction, PBL height, etc..

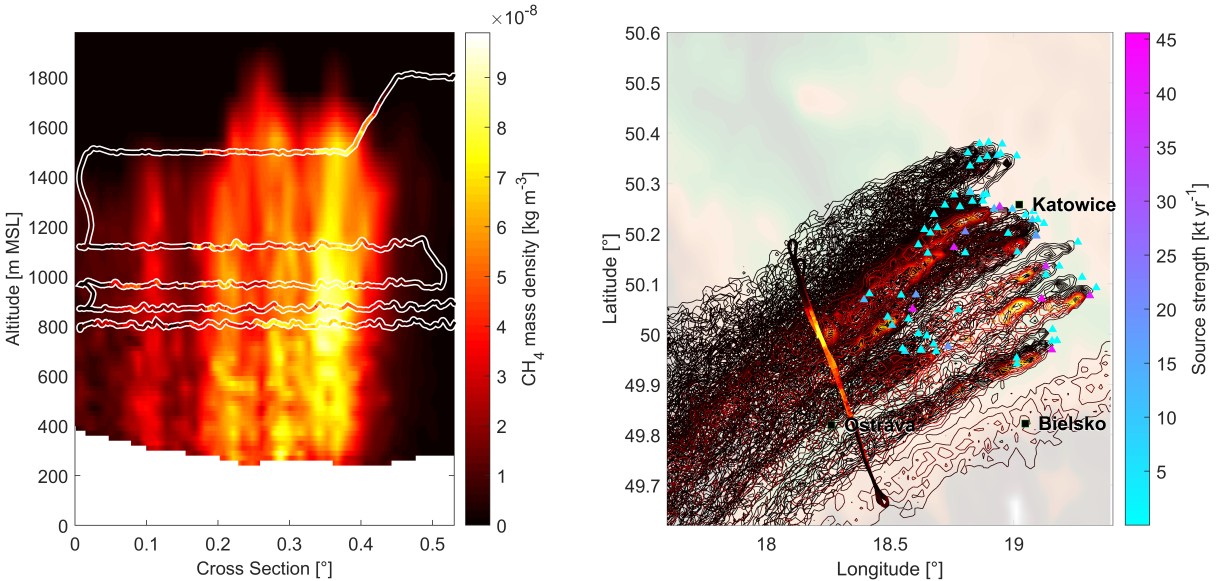

**Figure 8.** Left: Interpolated cross section of the model output along the downwind wall including scattered in situ observations of $CH_4$ for the afternoon flight on June 6th, 2018. The colorbar also applies to the isolines on the right. Right: Top-down view on the model output and the downwind wall observations $\rho$ at $750\,\mathrm{m}$ a.g.l. Triangles mark simulated emitters with colors corresponding to the optimized source strengths. Both panels show a snapshot of the model output at one fixed time chosen as the center time of the downwind wall phase.

    The total instantaneous emission estimate directly follows from the optimized parameters $x_i$ via Eq. 2. The emission estimate obtained for the morning flight on June 6th, 2018 using the model based approach amounts to $\Phi = \mathbf{452 \pm 78}\,\mathrm{kt\,yr^{-1}}$. It differs from the yearly averaged inventorial emission estimates for the USCB region by approximately **-37** % for EDGAR v4.3.2 and

**2** % for the E-PRTR inventory, respectively. The retrieval for the morning flight yields 44 degrees of freedom for signal and a total of 38 out of 74 modeled sources actively emitting. Here, the large amount of degrees of freedom for signal is indicative for the validity of the total emission estimate. The latter can be confirmed by the negligible impact of the a-priori $\sum (\mathbf{1} - \mathbf{A})\,\mathbf{x_a}$





versus the observations $\sum \mathbf{A}\mathbf{x}$ in the emission estimate. The algorithm makes use of the large number of modeled sources to enable a total emission estimate plus additional information on individual sources.

To enhance confidence in the emission estimate an afternoon flight of the DLR Cessna 208B was carried out a few hours after the morning flight ended on June 6th, 2018. Due to consistent wind directions on that day, the flight pattern was kept as close as possible to the morning flight. The flight trajectories for both flights are depicted in Fig. 1. Figure 9 shows the corresponding time series of the measured and modeled $CH_4$ mass density as a function of flight time during the downwind wall phase with source coefficients $x_i$ already optimized. Alike for the morning flight a good overall match between model and

in situ observations can be observed with a mean bias of $3 \times 10^{-10}\,\mathrm{kg\,m^{-3}}$ and a root mean square error of $1 \times 10^{-8}\,\mathrm{kg\,m^{-3}}$. The sensed mixing ratios are lower compared to the morning flight due to a further developed and hence more diluted boundary layer in the afternoon. The corresponding snapshot 2D scene is depicted in Fig. 8, with the left panel showing a cross section of the model output along the downwind wall including the in situ observations of $\rho$. The right panel of Fig. 8 shows the top-down view and the downwind wall observations as before. Both panels show a snapshot of the model output at the center time of

the downwind wall phase. It is evident from Fig. 8 that the inclined boundary layer height observed during the flight is nicely captured by the model. Boundary layer depth is generally enhanced compared to the morning flight. Plume trajectories are streamlined implying consistent winds over time.

The emission estimate obtained for the afternoon flight on June 6th, 2018 using the model based approach amounts to $\Phi = \mathbf{442 \pm 75}\,\mathrm{kt\,yr^{-1}}$. The obtained instantaneous emission estimate differs from the yearly averaged inventorial emission

values for the USCB region by approximately **-40** % for EDGAR v4.3.2 and **-2** % for the E-PRTR inventory, respectively. Estimated $CH_4$ fluxes are however consistently lower than EDGAR v4.3.2 inventorial data for both flights presented herein using the proposed method. The retrieval for the afternoon flight yields 40 degrees of freedom for signal and a total of 43 out of 74 modeled sources actively emitting. Both flights yield similar $d_s$ values, indicating that neither flight can be used on its own to retrieve all modeled sources.

**4.5   Single source apportionment**

The model based approach provides a unique advantage over established mass balance techniques in terms of spatial information, as it enables attributing sensed $CH_4$ mole fractions to remote sources at distances of tens to hundreds of kilometers. The total emission estimate has been introduced in Sect. 4.2 as the sum over $n$ sources $\varphi_i$ that are individually scaled with a coefficient $x_i$. The emission rate $\Phi_i$ corresponding to the i-th source is thus given by $x_i\,\varphi_i$. Here, the availability of data from

both research flights on June 6th, 2018 was exploited to estimate emission rates $\Phi_i$ for individual sources. By combining data from both flights the number of degrees of freedom for signal increases to $d_s = 48$ with 42 sources actively emitting.

Figure 10 illustrates $\Phi_i$ in $\mathrm{kt\,yr^{-1}}$ for all modeled emitters (mining shafts) taking into account both research flights on June 6th, 2018. The blue bars represent the estimated $\Phi_i$ and are to be related to the yearly average values (slim green bars) for each mining company reporting to E-PRTR 2017 for illustrative purposes. The estimated uncertainty depicted in Fig. 10

includes systematic uncertainties derived from a variational ensemble and statistic uncertainties due to the fit algorithm used (see Sect. 4.3). The variational ensemble introduced in Sect. 4.3 includes scaling coefficients $x_i$ subject to systematic variations





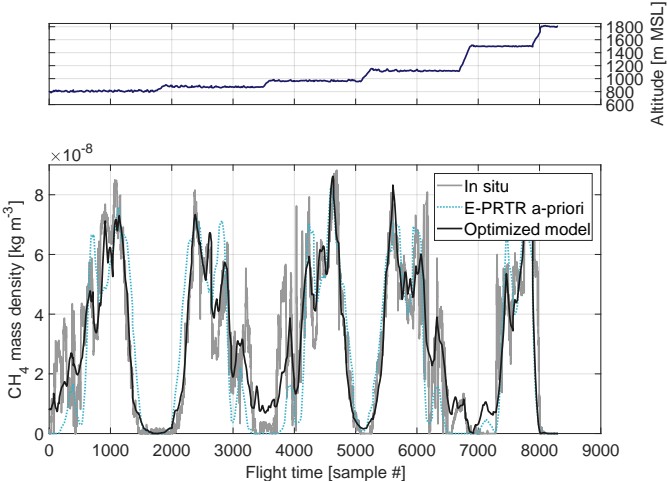

**Figure 9.** Time series of the in situ measured (gray) and modeled $CH_4$ mass density (solid black) with subtracted background as a function of flight time during the downwind wall phase of the afternoon flight on June 6th, 2018 with optimized source coefficients $x_i$. The dotted light blue line corresponds to the same forward simulation using scaling coefficients deduced from E-PRTR 2017.

in key sources of uncertainty. Systematic uncertainties for each source are directly obtained from this ensemble run. Differences in estimated and reported (E-PRTR 2017) $\Phi_i$ are evident. This is however expected due to the comparison of instantaneous emission estimates and yearly averages.

## 4.6 Uncertainty analysis

The influence of several variables on the total flux estimate $\Phi$ has been computed following Sect. 4.3. Figure 11 shows the influence of an error in wind speed ($\sigma_u = 0.9\,\mathrm{ms}^{-1}$), wind direction ($\sigma_d = 5\,°$), PBL height ($\sigma_{pbl} = 100\,\mathrm{m}$) and a source dislocation ($\sigma_{sd} = 1\,\mathrm{km}$) to total uncertainty for the flights detailed in the previous section. An assumed error in sensed mole fractions ($\sigma_c = 10\,\mathrm{ppb}$) is intended to include an error due to wrongly chosen background. The error on wind speed $\sigma_u$ is taken as the standard deviation of the difference between WRF modeled wind and non-assimilated in situ observations from the data depicted in Fig. 5. The same holds for the wind direction. The difference between modeled data and observations should therefore reflect overall uncertainty in these variables. Two spiral-up soundings out of the PBL revealed a boundary layer height of $1150\,\mathrm{m}$ at 0937 UTC and $1300\,\mathrm{m}$ at 1145 UTC. Based on these two soundings the uncertainty on boundary layer depth is estimated with $\sigma_{pbl} = 100\,\mathrm{m}$ for the downwind wall phase between 1000 UTC and 1100 UTC. For this sensitivity analysis, the WRF fields were perturbed systematically during the FLEXPART read phase in "readwind.f90". The source dislocation was implemented in the FLEXPART configuration file. It is evident from Fig. 11 that all selected error sources contribute on a similar level to total systematic uncertainty, which is ultimately computed as the standard deviation of the ensemble.

In addition to the derived systematic uncertainties, statistical errors related to the least squares fit have been computed following Sect. 4.3. Figure 12 depicts the Jacobian with respect to $x_i$ and the observations of the morning flight on June 6th,





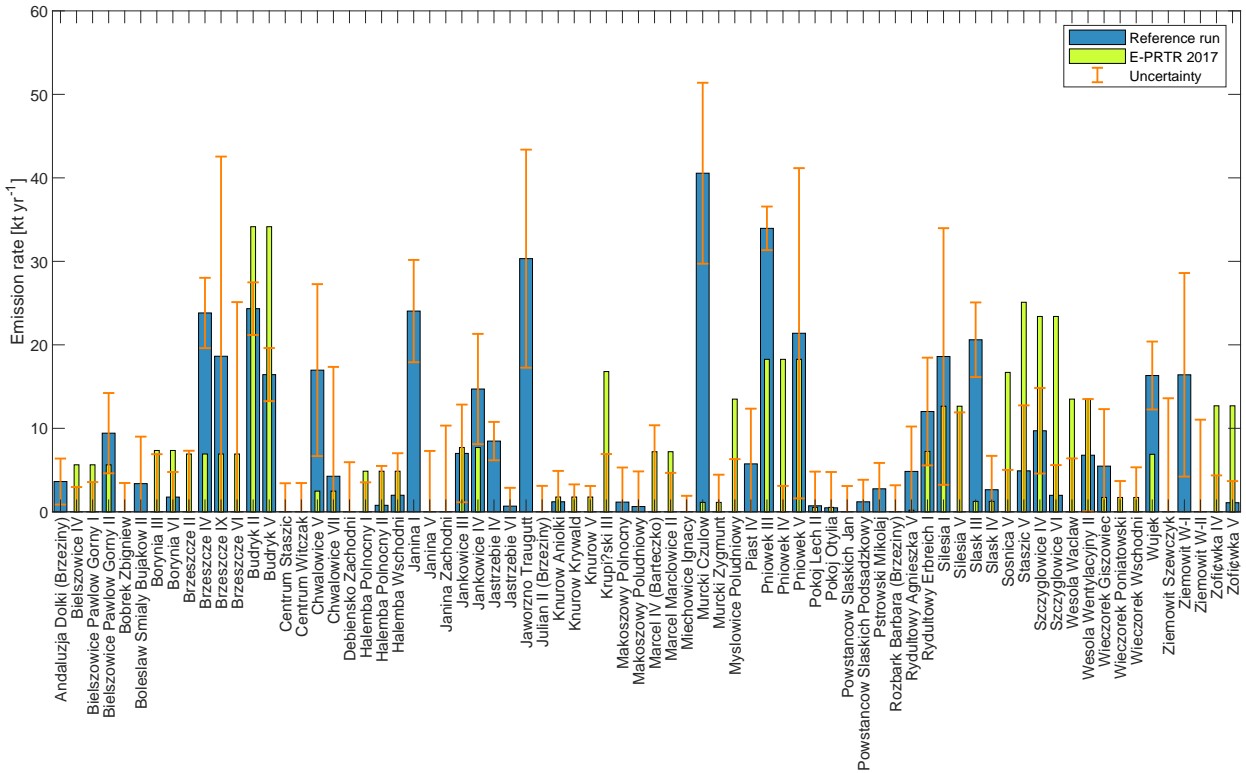

**Figure 10.** Emission estimates $\Phi_i$ (blue bars) in kt yr$^{-1}$ for 74 individual mining shafts using the morning and afternoon flight of June 6th, 2018. Slim green bars are the reported yearly average values for each mining company (E-PRTR 2017) evenly distributed among the respective ventilation shafts. The orange error bars stem from the quadrature sum of the statistical uncertainties $\epsilon_i$ (computed from the parameter covariance matrix $\hat{\mathbf{S}}$) and the uncertainties $\sigma_{ensemble}$ derived from a variational ensemble with systematically perturbed parameters.

2018. It describes the change in residuals introduced by a change in parameter $x_i$. From this figure, it can be seen that all 74 modeled sources were sampled by the aircraft using the chosen flight pattern, as all scaling coefficients are represented in the Jacobian. Scaling coefficients $x_i$ can therefore be deduced using a least squares fit. For the fluxes emanating from the USCB area, the statistical uncertainty $\epsilon_i$ computed from $\nabla_x J$ following Eqs. 7-8 amounts to approximately 34 kt yr$^{-1}$ or 7 % respectively.

Ultimately, the total uncertainty for the morning flight (same for the afternoon flight) on June 6th, 2018 is the sum of systematic (44 kt yr$^{-1}$) errors from the ensemble runs and statistical uncertainty (34 kt yr$^{-1}$) from the fitting algorithm adding up to approx. **17 %** relative uncertainty.





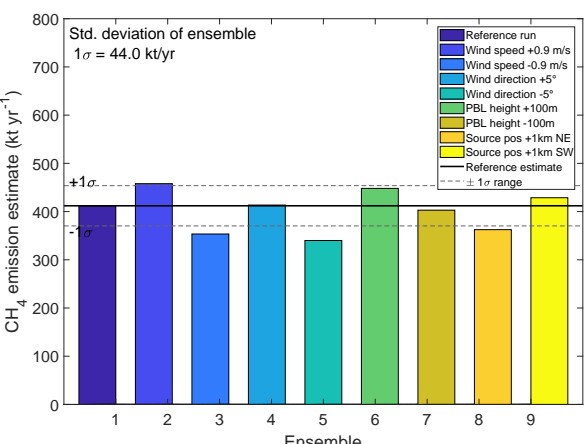

**Figure 11.** Ensemble runs to assess uncertainty in the flux estimates derived using a model based approach. All selected error sources contribute to total uncertainty on a similar level.

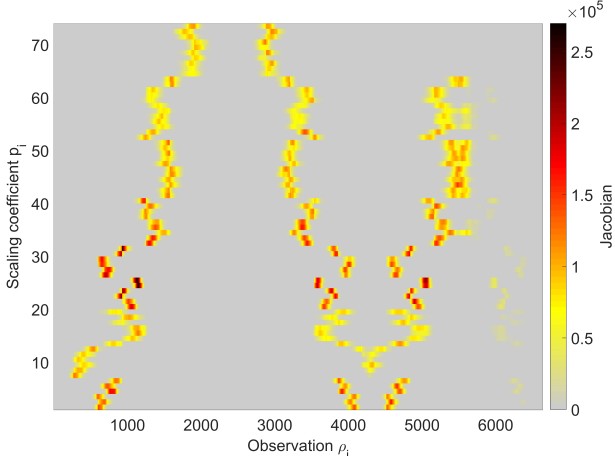

**Figure 12.** Jacobian with respect to $x_i$ and the observations $y_j$ for the morning flight on June 6th, 2018. All scaling coefficients $x_i$ are sensitive to variations in $y_j$ and can therefore be deduced from a MAP fit. Measurements centered around observation 2300 are not covered by the model and are thus obsolete for flux estimation using this particular flight.

## 5 Conclusions

A modified Aerodyne Dual QCLS instrument has been deployed aboard the DLR Cessna 208B in the context of the CoMet

1.0 campaign in early summer 2018 with the goal of estimating hard coal mine $CH_4$ emissions emanating from the USCB area - Europe's largest coal extraction region. Intensive mining activities and the heavy industry spread around the city of Katowice lead to significant amounts of GHGs emitted into the atmosphere. The reported inventorial $CH_4$ emission rates for





the entire USCB region amount to $720 \, \mathrm{kt \, yr^{-1}}$ (EDGAR v4.3.2) and $448 \, \mathrm{kt \, yr^{-1}}$ (E-PRTR 2017). The latter corresponds to $12.5 \, \mathrm{MtCO_2\text{-}eq \, yr^{-1}}$ using a $CH_4$-$GWP_{100}$=28 from the IPCC Fifth Assessment report (Pachauri et al., 2014). Assuming an

average carbon content of $75 \, \%$, a net calorific value of $29 \, \mathrm{MJ \, kg^{-1}}$, an emission factor of $94 \, \mathrm{tCO_2 \, (TJ)^{-1}}$ and the approximate 75 million tonnes of coal extracted from the USCB every year results in yearly $CO_2$ emissions of $205 \, \mathrm{MtCO_2 \, yr^{-1}}$ from burning of the extracted coal. The $CH_4$ emissions from mining alone therefore make up for approximately $6 \, \%$ in terms of GWP.

Estimates of coal mine $CH_4$ emissions in the USCB were derived using a model approach based on the Eulerian WRF
model and the Lagrangian particle dispersion model FLEXPART-WRF. Data assimilation further exploits the availability of additional data products, e.g. wind lidar soundings during the CoMet 1.0 campaign. Due to the known locations of the coal mine ventilation shafts, sources are modeled forward in time assuming a constant emission rate. Modeled data are then extracted at the aircraft positions in space and time and compared to actual airborne in situ observations. Here, meteorological driver data was generated using the WRF v4.0 model with continuous assimilated wind lidar soundings using WRF's OBS-FDDA
and WRFDA subsystems. After validation with unassimilated in situ measurements, data were fed into the Lagrangian particle dispersion model FLEXPART-WRF and used to model the exhaust plumes of the ventilation shafts. Using an inverse modeling approach, a-priori emission data from E-PRTR 2017 are optimized to allow a better fit to the observations. Thereby, total emission estimates for the USCB area of $\Phi = \mathbf{452 \pm 78} \, \mathrm{kt \, yr^{-1}}$ and $\Phi = \mathbf{442 \pm 75} \, \mathrm{kt \, yr^{-1}}$ were obtained for a morning flight and an afternoon flight on June 6th, 2018, respectively. Morning and afternoon flights differ by only $\sim 2 \, \%$ corresponding to
an excellent agreement well within the uncertainty range. The obtained emission estimate differs from the inventorial emission estimates by approximately -37 % / -40 % for the EDGAR v4.3.2 inventory (morning flight / afternoon flight) and $\pm \, 2 \, \%$ for the E-PRTR inventory, respectively. Differences in estimated and reported emission rates are however expected due to the comparison of instantaneous estimates and yearly averages. This is in line with previous studies hinting towards EDGAR v4.3.2 overestimating $CH_4$ emissions in the USCB (Luther et al. (2019); Fiehn et al. (2020)). Uncertainty estimates include systematic
contributions from ensemble runs and statistical uncertainty introduced by the fitting algorithm. Data from both research flights are further exploited to estimate individual source contributions. Differences between individual estimates and E-PRTR reported emissions are observed. This is expected due to several reasons: limited amount of measurements relative to the yearly averages provided in the inventories, wind directions do not differ by much between the two flights and the evenly distributed emissions among the ventilation shafts for each mining company. In general, the approach described herein delivers more
information compared to the conventional mass balance, albeit at increased effort: wind lidars need to be deployed during the measurement campaign, models need to be run, wind lidar data needs to be assimilated and inverse estimation techniques need to be applied. The additional possibility of remote source attribution however, coupled with the results obtained in Sect. 4.2 for the regional USCB anthropogenic $CH_4$ emissions make this approach a potent alternative to the mass balance technique. Although retrieving estimates for individual emitters is not possible using only single flights, due to sparse data availability, the
combination of two or more flights allows for exploiting different meteorological conditions to enhance confidence on facility level estimates.



*Code and data availability.* Data are available from the HALO-DB database https://halo-db.pa.op.dlr.de/. WRF v4.0 can be downloaded from https://www.mmm.ucar.edu/weather-research-and-forecasting-model. FLEXPART-WRF can be downloaded from https://www.flexpart.eu/. Model setups for both employed models are available upon request.

*Author contributions.* Anke Roiger and André Butz developed the research question. Anke Roiger, Julian Kostinek, Maximilian Eckl, Alina Fiehn and Andreas Luther took an active part in the CoMet field campaign by operating instrumentation, collecting and analyzing data. Norman Wildmann deployed wind lidars, collected and analyzed data. Andreas Stohl contributed substantially to the inverse method. Andreas Fix coordinated the CoMet field campaign operations. Theresa Klausner provided emission data for individual shafts. Christoph Knote provided and optimized the WRF model configuration. Julian Kostinek wrote the paper.

*Competing interests.* The authors are not aware of any competing interests.

*Acknowledgements.* We thank DLR VO-R for funding the young investigator research group "Greenhouse Gases". We also acknowledge funding from BMBF under project "AIRSPACE" (Grant-no. FKZ01LK170). We greatly appreciate continuous support from Hans Schlager and Markus Rapp from DLR. We thank Paul Stock, Monika Scheibe and Michael Lichtenstern from DLR for engineering support. Furthermore we would like to thank everyone involved during the CoMet field campaign for their relentless dedication and the helpful discussions,
especially Jarosław Necki and Justyna Swolkien who provided valuable information on $CH_4$ ventilation and took an active role as local advisers during the campaign period.





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
