# Peer review of "Estimating Upper Silesian coal mine methane emissions from airborne in situ observations and dispersion modeling"

_Atmospheric Chemistry and Physics, 2020_

## Referee Comment (RC1) · Anonymous Referee #2 · 21 Dec 2020

GENERAL COMMENT

The manuscript describes a study to estimate methane emissions from the Upper Silesian coal mining area using an air craft campaign. Two flights encircling the region have been conducted that measured methane concentrations up- and downwind in the plume. High resolution model simulations were used to relate the observations to possible emission locations, and in this way estimate the emissions for the chosen campaign day. Studies like this are highly relevant for climate research since the help to obtain insight in an important source of atmospheric methane, the second most important greenhouse gas after carbon dioxide. The subject is therefore well within the

scope of ACP, and could be published after some minor changes.

The used method is rigorous and robust. The description of the air craft flights is very informative, including the reasoning behind the flight plan. The simulations are based on combination of a trajectory model (FLEXPART) driven by a high-resolution meteorological model (WRF), where the later is also guided by wind lidar profiles taken during the same campaign. This gives trust that the simulations are as close to reality as can be expected from such simulations. Also the estimate of the emissions is done rigorous and seems hardly guided by a priori assumptions.

The results show that using such campaigns it is in principle possible to make a rather accurate estimate of actual emissions in a region with a large number of small sources that together aggregate into a major plume. Since the campaign is not more than a snap shot, it cannot be expected that these results can be extrapolated to for example a yearly total. However, could the authors give some reflection on how to use this kind method to do so? Could these campaigns be regularly repeated, or would for example regularly performed soundings (with FTIR or AirCore) be an alternative?

DETAILED COMMENT

Lines 80-86: Is there information available on how the EDGAR inventory estimated these CH4 emissions? In section 4.4 a comparison is made with E-PRTR. Since the later is more "specific", one could argue that it is more accurate. Aren't these numbers then not used in EDGAR?

Section 4.1: The WRF model assimilates wind profiles from the 3 lidars. This gives trust in the quality of the simulations, but how essential are these in the end? If these were not used, what could roughly be the change in the results?

Figure 5: Temperature is 2-4 degrees biased ($\sim$ 1%). Is that problematic for computing air densities etc? Think it should be related to the error estimates in section 4.3. Observed temperature is used in eq. 10 for the in-situ observations, should assimilated

temperatures then be used in the simulations?

Line 190: Where is the source index $i$ used in the emission rate $\varphi_i$ ? I guess that both emitted mass and emission time are source dependend.

Line 209: What do we learn from $d\_s$? Much more observations than sources, so in theory overdetermined problem. If $d\_s$ equals number of sources, is the estimate than exact?

Lines 239-240: This is a correct description of the Jacobian, but has that been used in the error estimates described in this section?

Figure 7, right panel: what is the background color, emissions from EDGAR ?

Line 280: Why not Fig 6 and Fig 9 as two panels next to each other?

Line 297. A scatter plot with x for afternoon vs morning would be useful to see if the algorithm estimates emissions to be present from the same locations, and how different these are.

Line 298. What is meant with "... neither flight can be used on its own ..." Here a remark could be placed that in the next section the morning and afternoon flights are analyzed together.

Section 4.6 The estimated emission totals have an uncertainty of 16% (std.dev.) Could this value be related in someway to a yearly total? For example, how many campaigns would be needed to come within an accuracy of say 2% over the year?

SPELL AND GRAMMER

line 11: no comma: "... flights due to ..."

line 25: ".. processing, and transport .."

---

## Short Comment (SC1) · 23 Dec 2020

The authors of the article titled "Estimating Upper Silesian coal mine methane emissions from airborne in situ observations and dispersion modelling" presented the unique results of the direct measurements of methane concentration in the air over the USCB region. The article is very interesting and aims at very important target: experimental verification of the methane emission estimate basing on momentary measurements – in this case airborne. For that purpose analyses performed with QCL/ICL instrument were coupled not only with mass balance technique (already successfully used and presented in publication Fiehn at al 2020 (https://doi.org/10.5194/acp-20-

12675-2020 ) but with the dispersion model. Beside estimates of total momentary emission, it allowed for a partition of the observed signal to particular ventilation shafts. This is the first attempt of such an initiative in case of USCB. Its hard work as the density of sources is very high and population density is also enormously large implying other sources like landfills and natural gas leakages taking part in the methane budget. In my opinion, the "wall pattern" methodology is not sensitive enough to scale correctly emissions in the group of sources lying in the radius of few km with an application of dispersion modelling. This would require further circle flying or at least more flying along other walls to obtain additional information reducing the large degree of freedom. Although I appreciate the acknowledgement of authors for providing advice on the ventilation shafts location, I would like to underline the role of uncertainty analysis in the validation of model results for such complex source structure as occurs over USCB. The discussion of the result with the rational data of shaft activities would increase the value of this article. My comments are related mostly with chapter 4.5 and figure 10, which in my opinion is the heart of the article. Before I will comment on the results for particular shafts, where there is a substantial difference between E_PRTR value and model estimation, I would like to divide the USCB region to the West and East part. There is not much comment for the results from a western part but for some reason, the eastern part makes the article very problematic. Authors are aware of this problem as they stated: "The discrepancy between model and observation at the lowermost (first in time) flight leg, corresponding to the southernmost trajectory section in Fig. 7 (right panel) can not be reproduced by any of the included emission sources. A possible source is urban CH4 emissions of Krakow, located to the east of the USCB region" Unfortunately, the hypothesis of Kraków city influence is unrealistic basing on the wind data presented (from 3 different location in USCB, where Windcubes were installed) and the angle between the city and flight route (larger than 65deg from the southernmost point– Fig1). However, if it happened due to the variation of wind direction or mass trajectories than the whole chapter 4.6 is not correctly calculated Wind dir and displacement distances uncertainties are underestimated. Figure 10

presents the emission fluxes of the shaft in order of their name. It would be much more informative for readers if the authors would place the emissions according to the position of the centre of plume encountered by an aircraft during flights. This would give them a possible explanation of the reason why the scaling factors optimized by the model is substantially biased for eastern UCSB. Comments for specific shaft emission estimates: Janina I ( 25kt/y - incorrectly) – the coal mine Janina is currently operated by Tauron Wydobycie company. It is the mine opened in 1907 but still containing one of the biggest profitable coal resources in Poland. Currently, there are 3 coalbeds excavated in this mine (at the relatively shallow level in comparison to other active coal mines) and none of them contains high methane amount. The highest (only one from 4 !)methane content of the coal is 0.2m3/MgCoal [table 20, https://www.tauron-wydobycie.pl/sites/default/files/Za%c5%82%c4%85cznik%20nr%201.5%20Janina-tekst1.pdf], what is 100 times less than the coalbeds in another mine operated by the same company (KWK Brzeszcze). This mine was always treated as "non-methane" mine from the methane explosion point of view and no methane incident was recorded throughout its history. Attribution of the wrong scaling factor is clear in this case. The possible reason why the model enhanced this direction compared to the measurement data lies in the existence of very big landfill in Balin county (approx. 7km NNE) which can release up to 10ktCH4 yearly. Brzeszcze VI (0kt/y - correctly) – This shaft was closed (filled with gravel and clay) in the year 2017 but has not been used since 2015, so the decomposition of yearly emission from the mine should include only 3 shafts (no II, IV and XI) not 4. This will increase the agreement in case of Reference run and E-PRTR database and in other databases where the emission is reported by this mine. Brzeszcze II (0kt/y - incorrectly)- All these shafts are located approx. 800m from each other looking perpendicular to wind axis (in reality II and IV are 1.9km apart, II and IX are 2.4 km apart). In this case, it is doubtful that model will attribute so drastically different scaling factors for this shafts (for II the estimated emission is 0 while for IV and IX it is close 20 kt/year) if it works correctly. I, personally measured emission by this shaft in the year 2018 at the comparable level with other two, however, Gaussian plume

model is also unreliable for such estimations – so I can't reveal it for comparison. Piast IV (5kt/y - unc. 100%)– this is strictly non-methane coal mine shaft, where the amount of VAM is negligible like another mine in the region (Ziemowit). For this reason, the result of scaling factor optimization is unreliable for both shafts (Piast IV and Ziemowit I (15kt/y - incorrectly)). Jaworzno-Traugutt (30kt/y – incorrectly) – Currently the shaft should be named Sobieski-Traugutt as the mine is under the control of Tauron Wydobycie company and was renamed to ZG Sobieski. The coalbeds excavated in this mine are methaneless, it is non-methane mine. The value proposed by the authors is exaggerated by few orders of magnitude as it would require a high content of methane in the coal, which is absolutely not a case here [e.g.: chapter 1.1.6.5, http://bip.katowice.rdos.gov.pl/files/obwieszczenia/53136/Obwieszczenie_RDOS_Katowice_WOOS_4235_5_2015_KC_2͞ Murcki Czułów (40kt/y – incorrectly) – Coal mine Murcki was connected with coal mine Staszic and subsequently closed. It is not operative now but its shafts are used for ventilation of part of the Staszic excavation works. The attribution of methane release to basically closed mine might, in this case, be reasonable. However, decision of the SRK (current owner of Murcki) and Staszic mine – the Czułów shaft has been closed in 2015 and since 2016 hasn't been used for ventilation purpose anymore (shaft was filled with gravel and clay) and finally in August 2018 it was dismounted [https://nettg.pl/news/152132/gornictwo-runela-wieza-szybu-czulow]. It is not possible that this amount of methane would be released by this shaft in June. So, the model has optimized the scaling factor wrongly, and surprisingly this was the highest methane emission rate attributed to any of the shafts. Murcki Zygmunt (0kt/y – incorrectly)– In opposition to shaft Czułów, this shaft is still operating and took all the ventilation work in relation to new longwalls work by Staszic mine in one of the deepest coalbeds containing a large amount of methane. In this case, however, the model has attributed scaling factor to zero with no methane emission. Comments regarding few westerly located shafts: Pniówek IV (0kt/y – incorrectly)– The active shaft belonging to one of the biggest methane emitting mines over USCB. Direct measurements recorded in the same time when the flight was performed allow to confirm that this shaft was operating

and releasing VAM (methane associated with ventilation) on usual level (approx. 0.2% CH4 with 10000m3/min of air). For an unknown reason, the scaling factor reduced this emission to negligible values. Chwalowice V (17kt/y – might be correct) – The shaft belongs to the mine with two levels of exploitation. Shaft V was renovated in the year 2009 due to the demands of a new coalbed being planned for excavation. For this reason the shaft release much more methane than shaft no.VII which ventilates the older coalbeds containing less methane. Here, the proximity of the shafts to flight route allowed for the rational application of optimized scaling factors. However, methane emission of the mine is at the lower or same level as Marcel, Rydultowy mines, where deeper coalbeds are excavated (contain more methane). Summarizing, in the eastern part of USCB there is a mismatch of emission attributed to particular shafts at the summary level of 100Kt/y with the biggest value 40kt/y for the single shaft. I agree with authors that the momentary value should not be directly compared with yearly release reported by the coal mines as the day to day (or even hour to hour) emission my change by 100% (e.g. increase from 10 to 20 m3/min). For this reason, all the modelled values should be expressed in short time units (minutes or hours). But attributing large emission to a closed shaft or non-methane shafts are beyond of such explanation. In my opinion, the authors overestimate the accuracy of the dispersion model by an order of magnitude. Obviously, it has not enough spatial resolution to distinguish between particular shaft in case of single-wall pattern measurement. I would like to give the appreciation for the authors to organize the COMET campaign and invest a lot of time to obtain great and novelty results which certainly are valuable material for publication. Jarek Necki (AGH –Univeristy, Krakow, Poland)

---

## Referee Comment (RC2) · Anonymous Referee #1 · 5 Feb 2021

The manuscript by Kostinek reports an interesting case study of top-down greenhouse gas emission quantification on the region scale using airborne observations, sophisticated atmospheric transport simulations and an inverse modelling framework. They estimate the CH4 emissions from coal mining shafts in Upper Silesia, one of the CH4 emission hot spots in Europe. The research topic is very relevant regarding greenhouse gas emission mitigation strategies under the Paris Agreement and as such deserves publication. The study design, the applied transport simulations and partly the inverse modelling approach are sound and mostly presented well in the manuscript. The authors took great care to assure adequate atmospheric transport simulations and made an important effort to characterise the connected uncertainties, adding some

valuable new concepts to the field. There are two major concerns, 1) regarding the feasibility of assigning emissions to individual ventilation shafts given the flight observations taken at considerable distance from individual sources and 2) concerning the omission of CH4 sources other than coal mines. These issues are detailed below and an answer will probably require additional analysis and revisions of the manuscript. However, I would encourage the authors to address these additional points, so that their very valuable analysis can be published in ACP.

Major comments

Inverse modelling method: I have two major concerns with the emission estimation method.

1) Emission attribution to individual shafts

The first concern is the attribution of emissions to individual facilities and shafts. Looking at figure 7 and 8 but also at the names of the shafts in figure 10, it is clear that many of the individual shafts cluster around individual mines at distances not much more than a kilometer. Furthermore, a lot of the locations actually line up with the main wind direction. In this situation it seems to be virtually impossible to estimate emissions from individual shafts from the presented observational data which was only taken at a single downwind curtain. The results presented in figure 10 are most likely a fine example of overfitting the observations and obtaining a 'noisy' a posteriori result. The problem is also apparent from figure 11, which seems to indicate that although there is some sensitivity to all emission shafts, sensitivity is much larger for certain shafts than for others. Given the observational data, the problem cannot be overcome in a general way, but at least the covariances in the inversion should be designed in a way that will limit overfitting and the overinterpretation of emission results. I would suggest that the authors modify the design of their a priori covariance. Currently, they only include diagonal elements. Hence, they assume uncorrelated uncertainties even for shafts from the same facility. I think it would be useful to explore by how much the results change if

correlated a priori uncertainties for shafts from the same facility and/or shafts a shorter distances would be introduced (positive off-diagonal values in the a priori covariance matrix). Furthermore, the a posteriori covariance matrix should be explored in order to see if many of the emitters actually show negative covariances to one another. This would indicate that there remained a large uncertainty as to which shaft the emissions had to be assigned. In general, the manuscript should better highlight that the uncertainty on the shaft-level remained large and that the observational data is to limited for a more specific estimate.

In this context, it would also be good to present the shaft level emissions on a map and compare with spatial inventories.

This issue was also raised in the comment by J. Necki and he provides valuable discussion on a per shaft basis that should not be ignored in a revision of the manuscript.

2) Neglect of other than coal mine emissions

In section 2 it is discussed that EDGAR assigns about 14 % of total CH4 emissions in the area to non-coal mine emissions. The authors largely ignore these emissions in their analysis. Only in section 4.4 the possibility of emissions from the city of Krakow are discussed but ultimately these were also not included for the inverse modelling. Although 14 % may not seem a very large fraction and one could argue that any kind of mis-attribution is covered by the uncertainty estimates, it will still strongly depend on the distribution of these missing emissions relative to the sampling locations. Their impact on the concentration observations may well have been much larger if sources were closer to the flight track than the coal mine emissions. Since the authors also find that total emissions for the region are considerably smaller than what was reported in EDGAR, the fraction of non-coal may also be larger than in EDGAR. Seasonality of emissions may also play a role here. Since this is early summer, temperatures were above annual average, possibly leading to larger than annual average emissions from microbial sources like landfills, waste water treatment, manure management, etc.

There seems to be a large discrepancy with EDGAR in terms of shafts and emission locations as well (red grid cell in the lower left corner of the encircled area close to Ostrava). Are these actually coal mine emissions in EDGAR? Looking at the area, one can also see a larger reservoir north-west of Bielsko Biala. Could natural emissions from this reservoir also play a role in the poorer agreement at the eastern end of the curtain flight? I strongly suggest that the authors reconsider their neglect of the non-coal mine emissions. They should obtain inventory data of these emissions, possibly not just from EDGAR (see comment below on inventories) but other more resolved inventories or from local information (see comment by J. Necki). With these emissions another FLEXPART forward run should be conducted and the resulting concentration either be removed from the observation vector before the inversion or an additional scaling factor for non-coal mine emissions should be introduced in the inversion.

Minor comments

p4: Regarding the use of the EDGAR inventory I would like to question if this is really the best available bottom-up inventory for the area. First of all, there is newer version of EDGAR available (v_5.0_GHG), which explicitly lists CH4 from coal exploitation as a separate category and is available for a more recent year (2015) than EDGAR 4.3.2. Furthermore and as part of the EU project CHE, TNO has compiled higher resolution (6 km x 6 km) inventories for Europe. They may be better suited than EDGAR (see https://www.che-project.eu/sites/default/files/2019-01/CHE-D2-3-V1-0.pdf; data usually available on request). This is not only important for the final comparison of obtained emission estimates but also relates to the question if and how non-coal emission need to be treated in the inversion framework.

p5, l105ff: A detailed description of the flight data is given here. Some of it is depicted as well in figure 5, but it would be valuable to present all recorded time series of meteorological and CH4 data along with flight parameters in a separate plot (supplement) highlighted by flight segment. One plot for each of the flights. If such a plot was already published previously a reference would be sufficient too.

p5, l109: Here it is mentioned that an upwind concentration is subtracted from the downwind measurements. Later on a different method for background subtraction is described. What was really used?

p5, l113: Why is detrainment/entrainment important to this study? The FLEXPART-WRF simulations don't exclude detrainment/entrainment processes or PBL growth. Detrainment/entrainment would be more of an issue for a mass balance approach.

p5, l117: While an estimate of the morning PBL height is given, its height is not mentioned for the afternoon flight. Please add. Maybe also comment on the growth of the PBL height between the two flights and how this relates to the question of detrainment/entrainment.

p7, L148f: Were the Doppler soundings the only observations that were nudged? What about other standard synoptic observations in the area?

p8, l158ff: Does this mean that 3Dvar and nudging were applied to the same observational data? That would not make sense in my view as the same information gets used twice. Rather use 3Dvar with smaller error covariance if the pull of the observations seemed too weak and such smaller uncertainties could be justified. Also, how were observational error covariances determined exactly?

p8, l167: Are these numbers the root mean square errors between model and observations for the 1-Hz sampling?

p8, l171f: How can this apparent offset in pressure be explained? Difficult to believe that the models (WRF nested in GFS) are off by that much, especially since the wind seems to match very well. Was there any comparison to other surface pressure data? Concerning the temperature offset: Does this vanish when you calculate potential temperatures? And same question as for pressure: were there any ground based measurements to compare to?

p9, inversion method: I got confused by the description here. First, a non-regularized

least square equation is presented for flux optimisation (eq. 1). Then regularization using a priori information and Bayes' theorem is advocated. To my understanding the resulting equations 4 and 5 only require a simple matrix inversion for solving for the a posterioir state. However, from line 211 onwards the application of a non-negative least square solver is presented. The latter is probably applied to equation 1, yielding a positive solution for x. However, if this was the case, I don't see why further analytical solutions to the cost function are presented in 4 and 5. I assume I am missing an important point here and would like the authors to clarify. If only a positively constrained solution for equation 1 is obtained I would think the results are even more overfitted as already mentioned above, since no additional a priori constraint on the individual sources would have been used. The description in the results section strongly suggests that this was the case. In equation, 4 I also think the last term Kx should also be Kx_a instead (see Tarantola eq. 3.37 or Jacob eq. 23).

p16, l336: Here the total uncertainty of the emission estimate is presented as the sum of the 'systematic' uncertainty (which I assume results from the a posteriori covariance; eq. 8) and the spread obtained from the sensitivity simulations. Why are these uncertainty terms not added quadratically?

Section 4.3: The way the covariance matrices for the a priori and the observation/model error are constructed most likely oversimplifies the true nature of the involved covariances and may lead to overfitted results. First, and already mentioned in the main comment above, the a priori covariance should acknowledge the fact that the a priori emission uncertainties will be correlated. This is true for shafts belonging to the same mining complex but may also be true for spatial distances between shafts. As mentioned above, I would suggest introducing off-diagonal elements in the covariance matrix to honor this fact. This would certainly lead to a smoothing out of the emissions across different shafts but is a more realistic approach. Furthermore, the observation/model covariance does not include off-diagonal elements either. However, the 1-Hz observations are certainly not independent from each other since they contain

tempo-spatial autocorrelation. The latter will also be present in 1-Hz model residuals. I would suggest to explore this auto-correlation in the residuals and add a temporal correlation length to the observation/model matrix accordingly. Adding these off-diagonal elements will probably reduce the impact of the observations on the a posteriori results, reflecting that they are not really independent from each other. Another way to get rid of the autocorrelation would be temporal averaging of the observations before using them in the inversion. This has its merits as well as it would also bring the spatial resolution of the observations closer to those of the transport model.

p11, l231: How was the transport model uncertainty estimated concretely? As the standard deviation of simulated concentrations from the 8 ensemble members?

p12, l248: Why not use the measurements from the upstream flight segment as background? The comparison with the model output seems to indicate that the overall plume was wider than the flight segments.

p14, l296: This sentence largely repeats the result from the previous sentence (EDGAR being much larger than the current estimate).

section 4.6: After reading the first few sentences, it was not clear to me how an uncertainty quantified by sigma was adopted in the transport model. I guess figure 11 makes it clear that 8 sensitivity runs were done where the respective variables were perturbed in one or the other direction globally, but not perturbed by random noise with the given sigma width. This should be made a bit clearer from the beginning.

p15, l325: If I understand correctly, original horizontal wind speeds as output from WRF were increased/decreased by 0.9 m/s. In doing so, the local mass balance of the wind field may well be destroyed as vertical wind speeds were not adjusted (correct?). This may lead to errors in the transport description of the LPDM. Have you given this any thoughts? Probably the impact was not to large and since this is only presented as a sensitivity case it is of less importance, but it may have lead to larger discrepancies from the reference run than anticipated. A similar question for the PBL height. Is the

latter taken from WRF or is the diagnostic calculation taken from FLEXPART? When increasing the PBL height just in FLEXPART vertical mixing in FLEXPART may then bring model particles to altitudes that in WRF are not part of the PBL and as such may have a distinctly different flow direction as flow in the PBL. As a consequence the differences to reference run may be larger than in a case where WRF PBL heights were larger/smaller. Hence, your change in the PBL height may give a slightly more pessimistic (larger) uncertainty.

p16, l23: Here it is mentioned that the statistical uncertainty was estimated from eq. 7 and 8 and it is referred to elements $e\_i$, which are the diagonal elements of the a posteriori covariance matrix. What about the off-diagonal elements of this matrix? Were the taken into account for the total uncertainty estimate?

Figure 12 and use of Jacobian: If I understand correctly, what is shown in figure 12 is the matrix K containing the elements $dy\_j/dx\_i$. However, the term Jacobian is also used in the manuscript for $grad(J(x))$. But J is not a vector-valued function and as such $grad(J)$ is not a Jacobian. Please clarify.

Technical comments

p2, l45. Karion et al. is missing a publication date.

Figure 1: The line indicating the afternoon flight is more orange than red (as described in caption). It looks like the map is showing total EDGAR emissions. How does the distribution of non-coal mine emissions look like?

Figure 3: Please label the WRF domain in the figure according to their definition in the text.

Figure 4+5: Please use the same colors for the different WRF runs. The legend is fairly small in both cases and needs to be enlarged, possibly put to the right of the sub-panels.

[Figure]

---

## Editor Comment (EC1) · Ilse Aben (Editor) · 7 Feb 2021

7 Feb 2021

Dear authors

Please make sure you address all points from all reviewers in your response and subsequent revised version of your manuscript.

In particular, please note that one of the official reviewers has raised some serious issues that I believe require your fullest attention in your response and the necessary adjustments in a possible revised version of your manuscript. Also the review submitted

by Jaroslaw Necki deserves the necessary attention.

Best regards, Ilse Aben (co-editor for the manuscript)
* * *

---

## Author Comment (AC1) · 17 Mar 2021

**In response to Anonymous Referee #2 comments from December the 21st, 2020.**

*The manuscript describes a study to estimate methane emissions from the Upper Silesian coal mining area using an aircraft campaign. Two flights encircling the region have been conducted that measured methane concentrations up- and downwind in the plume. High resolution model simulations were used to relate the observations to possible emission locations, and in this*
5  *way estimate the emissions for the chosen campaign day. Studies like this are highly relevant for climate research since the help to obtain insight in an important source of atmospheric methane, the second most important greenhouse gas after carbon dioxide. The subject is therefore well within the scope of ACP, and could be published after some minor changes.*

*The used method is rigorous and robust. The description of the aircraft flights is very informative, including the reasoning behind the flight plan. The simulations are based on combination of a trajectory model (FLEXPART) driven by a high-resolution*
10  *meteorological model (WRF), where the later is also guided by wind lidar profiles taken during the same campaign. This gives trust that the simulations are as close to reality as can be expected from such simulations. Also the estimate of the emissions is done rigorous and seems hardly guided by a priori assumptions.*

*The results show that using such campaigns it is in principle possible to make a rather accurate estimate of actual emissions in a region with a large number of small sources that together aggregate into a major plume. Since the campaign is not more*
15  *than a snap shot, it cannot be expected that these results can be extrapolated to for example a yearly total. However, could the authors give some reflection on how to use this kind method to do so? Could these campaigns be regularly repeated, or would for example regularly performed soundings (with FTIR or AirCore) be an alternative?*

Dear Referee,
Thank you very much for your kind and helpful comments on the analysis presented in the manuscript. We absolutely agree,
20  that extrapolation to yearly totals requires not only snapshot observations, but continuous measurements at various meteorological conditions. This manuscript is intended as a further step towards this goal. Achieving this goal will benefit substantially from concurrent FTIR soundings, aswell as active AirCores, as the method presented herein is not restricted to in situ data but can also be adapted to total column measurements. Total column measurements, especially mobile FTIRs, are in fact complementary to in situ observations and have already been used in explorative studies similar to the one presented here (Luther et
25  al., 2019).

1. **Lines 80-86: Is there information available on how the EDGAR inventory estimated these CH4 emissions? In section 4.4 a comparison is made with E-PRTR. Since the later is more "specific", one could argue that it is more accurate. Aren't these numbers then not used in EDGAR?**
   EDGAR v4.3.2 includes data from EPRTRv4.2 for the industrial sector, according to Janssens-Maenhout et al., 2017.
30   However, the same reference also states: "A point-source database, such as the European Pollutant Release and Transfer Register (EPRTR18), allows a more homogeneous input for an inventory compiled under such a facility-based approach. The European study of Theloke et al. (2011), which aimed to complement EPRTR point sources with information on diffusive sources per country that together match national totals, revealed large inconsistencies, which prevented closing the two approaches in a satisfying way." It seems like discrepancies between the two inventories are still large.

35  2. **Figure 5: Temperature is 2-4 degrees biased ($\sim 1\,\%$). Is that problematic for computing air densities etc? Think it should be related to the error estimates in section 4.3. Observed temperature is used in eq. 10 for the in-situ observations, should assimilated temperatures then be used in the simulations?**
   The in-situ concentration measurements use the actually measured temperatures, so the bias does not affect the measurements themselves but only the comparison between model and simulation. It is de facto related to the uncertainty
40   analysis. From our earlier studies we do know, that a systematic error in static air temperature manifests approximately an order of magnitude lower in emission estimates compared to errors in wind speed and direction. Nevertheless we included an error in sensed mole fractions of $\sigma_c = \pm 10\,ppb$ in the uncertainty analysis in Sect. 4.6. In theory - yes - it would be good to use assimilated temperatures in the simulations. Using data from the aircraft alone for the assimilation, however, does not improve on this bias in the present case. We tried different setups but did not succeed in reducing this

bias to a satisfactory level. Unfortunately the Doppler lidars can only report wind speed, direction at specific altitudes and derived quantities. It would certainly be great if these devices would be able to remotely sound air temperature in the future, in order to have a solid basis for temperature assimilation.

3. **Line 190: Where is the source index $i$ used in the emission rate $\varphi_i$ ? I guess that both emitted mass and emission time are source dependend.**
   That is correct. The relevant sentence has been revised to: "[...] A scaling coefficient $x_i$ is assigned to each of the $n$ sources $\varphi_i = m_{e,i}\tau_{e,i}^{-1}$, with the total emission time $\tau_{e,i}$ in seconds and the total mass emitted $m_{e,i}$ in kg for each simulated source. [...]"

4. **Line 209: What do we learn from $d_s$? Much more observations than sources, so in theory overdetermined problem. If $d_s$ equals number of sources, is the estimate than exact?**
   The problem is mathematically overdetermined, hence no exact solution exists. An approximate solution has to be found via e.g. a least-squares approach. Here we did not use normal least-squares but further included known information on the measurement uncertainties and a-priori knowledge on the sources. The degrees of freedom for signal $d_s$ describe the reduction in the normalized error on $\mathbf{x}$ introduced by the available observations and hence provides a measure for the improvement in knowledge of $\mathbf{x}$, relative to the a-priori, due to the observations. If $d_s$ equals the number of sources (unknowns in the system of equations), the a-priori has no influence on the retrieval. In that case all "information" originates from the measurements alone. However, it does not mean the estimate is exact, as the system of equations used may (and will) only partially reflect the "true" atmospheric "system of equations" (the truth) and since measurement errors will propagate heavily into the estimate.

5. **Lines 239-240: This is a correct description of the Jacobian, but has that been used in the error estimates described in this section?**
   The Jacobian is not relevant in this section of the manuscript, yet is used in Sect. 4.6. The intention was to introduce the quantities needed later to allow for easier reading. As it is not relevant in this section we removed the sentence in a revised version of the manuscript.

6. **Figure 7, right panel: what is the background color, emissions from EDGAR ?**
   The background colors from the right panels in Fig. 7 and Fig. 8 correspond to the topography in the USCB region. To the south the terrain becomes elevated as approaching the Tatra mountains. The $CH_4$ plumes are nicely advected into the Moravian gate. We revised both figure captions to include this info: "[...] Right: Top-down view on the model output and the downwind wall observations $\rho$ at $750\,\mathrm{m\,a.g.l}$ with underlaid topography. [...]"

7. **Line 280: Why not Fig 6 and Fig 9 as two panels next to each other?**
   Our intention was to treat both flights separately, before discussing the combined flights in Sect. 4.5. We do however agree, that the figures are similar and could be merged into a 2-panel figure for a better flow. We merged Fig. 6 and Fig. 9 in a revised version of this manuscript.

8. **Line 297. A scatter plot with x for afternoon vs morning would be useful to see if the algorithm estimates emissions to be present from the same locations, and how different these are.**
   Subregional emission estimates require at least two flights associated with different wind conditions. During single flights several emission sources are masked by sources closer to the point of measurement. In the present scenario, the large number of sources enables regional estimates for single flights, albeit subregional estimates will vary significantly and are not reliable for the reason mentioned above.

9. **Line 298. What is meant with "... neither flight can be used on its own ..." Here a remark could be placed that in the next section the morning and afternoon flights are analyzed together.**
   The degrees of freedom for signal of both flights are very similar (43 for the morning flight and 44 for the afternoon flight). A value of $d_s \sim 40$ indicates, that not all "information" stems from observations, or synonymously - the emission estimate is to some extent influenced by the a-priori. This is due to some sources being "obscured" by sources closer to

the point of measurement or sparse data availability. Enhancing the degrees of freedom for signal requires measuring downwind at different wind directions. Of course it would be convenient to sample at very different advection scenarios. This is however hardly possible, as one of the principal assumptions - constant emission rate over the timespan of the flight(s) - would be violated due to daily changes in ventilation volume of the mines. As a compromise, two flights on the same day with only slightly different wind conditions have been used in an effort to maximize likelihood of a constant emission rate. We rephrased and added the following clarifying sentence to the revised manuscript: "[...] Both flights yield similar $d_s$ values, indicating that not all information stems from observations alone. Hence, neither flight can be used alone to retrieve all modeled sources. In an effort to minimize the dependency on the a-priori, both flights will be analyzed together in the next section. [...]"

10. **Section 4.6 The estimated emission totals have an uncertainty of 16 % (std.dev.) Could this value be related in someway to a yearly total? For example, how many campaigns would be needed to come within an accuracy of say 2 % over the year?**
As you mentioned in the general comments above it would be possible to repeat these measurements under different meteorological, most importantly wind conditions, in order to minimize statistical uncertainty. This would however not improve on the systematic uncertainty. Consequently it is hard to come up with an answer on how many campaigns would be needed to reach an uncertainty of 2%.

11. **Spell and grammar**
Spell and grammar has been corrected.

---

## Author Comment (AC2) · 17 Mar 2021

**In response to Jaroslaw Necki's comments from December the 23rd, 2020.**

Dear Jaroslaw,
We greatly appreciate your very valuable comment on this manuscript. We will reply inline:

5    *The authors of the article titled "Estimating Upper Silesian coal mine methane emissions from airborne in situ observations and dispersion modelling" presented the unique results of the direct measurements of methane concentration in the air over the USCB region. The article is very interesting and aims at very important target: experimental verification of the methane emission estimate basing on momentary measurements – in this case airborne. For that purpose analyses performed with QCL/ICL instrument were coupled not only with mass balance technique (already successfully used and presented in publication Fiehn*

10    *at al 2020 (https://doi.org/10.5194/acp-20-12675-2020 ) but with the dispersion model. Beside estimates of total momentary emission, it allowed for a partition of the observed signal to particular ventilation shafts. This is the first attempt of such an initiative in case of USCB. Its hard work as the density of sources is very high and population density is also enormously large implying other sources like landfills and natural gas leakages taking part in the methane budget. In my opinion, the "wall pattern" methodology is not sensitive enough to scale correctly emissions in the group of sources lying in the radius of few km*

15    *with an application of dispersion modelling. This would require further circle flying or at least more flying along other walls to obtain additional information reducing the large degree of freedom.*

   As stated in Sect. 3 the primary goal of this mission was to obtain regional emission estimates from the entire USCB region. We do agree, that encircling individual shafts might be an alternative approach towards estimating single shaft emissions, albeit

20    only under certain conditions, that could not be met in the USCB region. It is mainly the lowest flight altitude and the required sharp turns, that hinders such an attempt, as the plumes are not sufficiently buoyant to reach the prescribed minimum flight altitude in the shafts vicinity. Furthermore, the plumes are not well-mixed and turbulence has to be taken into account close to the ventilation shafts, which certainly makes it even more difficult to obtain reliable emission estimates.

   *Although I appreciate the acknowledgement of authors for providing advice on the ventilation shafts location, I would like to*

25    *underline the role of uncertainty analysis in the validation of model results for such complex source structure as occurs over USCB. The discussion of the result with the rational data of shaft activities would increase the value of this article.*

   We do agree with your statement on the importance of an adequate uncertainty analysis. In fact we tried to be as rigorous as possible with the uncertainty analysis, as described in Sect. 4.3 and Sect. 4.6.

30    *My comments are related mostly with chapter 4.5 and figure 10, which in my opinion is the heart of the article.*

   In our opinion, the value of this article lies in the method described, as well as the results for the entire USCB region. Obtaining estimates from individual shafts from these large area flights is certainly an added value, that should be focused upon in upcoming studies. As stated in our reply to Referee #1, we concede that section 4.5 of the original manuscript might

35    have been misleading the reader to think that we can accurately estimate individual shafts from two downwind walls. We rephrased in a way to make clearer that - while formally reporting individual shaft emissions - there is large uncertainties and correlations among the emission estimates.

   *Before I will comment on the results for particular shafts, where there is a substantial difference between E-PRTR value and model estimation, I would like to divide the USCB region to the West and East part. There is not much comment for the results*

40    *from a western part but for some reason, the eastern part makes the article very problematic. Authors are aware of this problem as they stated: "The discrepancy between model and observation at the lowermost (first in time) flight leg, corresponding to the southernmost trajectory section in Fig. 7 (right panel) can not be reproduced by any of the included emission sources. A possible source is urban CH4 emissions of Krakow, located to the east of the USCB region" Unfortunately, the hypothesis of Kraków city influence is unrealistic basing on the wind data presented (from 3 different location in USCB, where Windcubes*

*were installed) and the angle between the city and flight route (larger than 65deg from the southernmost point– Fig1). However, if it happened due to the variation of wind direction or mass trajectories than the whole chapter 4.6 is not correctly calculated Wind dir and displacement distances uncertainties are underestimated.*

5     Using a particle dispersion model driven by a high-resolution meteorological model that assimilates three local wind-profiles from Doppler lidar measurements is among the best one can do in terms of calculating plume dispersion. Variations in wind direction and speed are de facto taken into account in the simulations, but not so in other methods, e.g. mass balance, Gaussian plume model. We also believe that our uncertainty analysis is representative of the actual uncertainties in the meteorological parameters.

10    *Figure 10 presents the emission fluxes of the shaft in order of their name. It would be much more informative for readers if the authors would place the emissions according to the position of the centre of plume encountered by an aircraft during flights. This would give them a possible explanation of the reason why the scaling factors optimized by the model is substantially biased for eastern UCSB.*

15    We considered other options for plotting Fig. 10 (ordered by name, emission rate, position of encounter) but after discussion with the team we decided to keep the ordering by name.

*Comments for specific shaft emission estimates: Janina I ( 25kt/y - incorrectly) – the coal mine Janina is currently operated by Tauron Wydobycie company. It is the mine opened in 1907 but still containing one of the biggest profitable coal resources in Poland. Currently, there are 3 coalbeds excavated in this mine (at the relatively shallow level in comparison to*
20 *other active coal mines) and none of them contains high methane amount. The highest (only one from 4 !)methane content of the coal is $0.2m3/MgCoal$ [table 20, https://www.tauronwydobycie.pl/sites/default/files/Za%c5%82%c4%85cznik%20nr% 201.5%20Janinatekst1.pdf ], what is 100 times less than the coalbeds in another mine operated by the same company (KWK Brzeszcze). This mine was always treated as "non-methane" mine from the methane explosion point of view and no methane incident was recorded throughout its history. Attribution of the wrong scaling factor is clear in this case. The possible reason*
25 *why the model enhanced this direction compared to the measurement data lies in the existence of very big landfill in Balin county (approx. 7km NNE) which can release up to 10ktCH4 yearly.*

    This is interesting information that could explain this attribution to Janina I, as the a-priori is also very low in that case. Thank you for pointing out the presence of a big landfill in this region.

30 *Brzeszcze VI (0kt/y - correctly) – This shaft was closed (filled with gravel and clay) in the year 2017 but has not been used since 2015, so the decomposition of yearly emission from the mine should include only 3 shafts (no II, IV and XI) not 4. This will increase the agreement in case of Reference run and E-PRTR database and in other databases where the emission is reported by this mine. Brzeszcze II (0kt/y - incorrectly)- All these shafts are located approx. 800m from each other looking perpendicular to wind axis (in reality II and IV are 1.9km apart, II and IX are 2.4 km apart). In this case, it is doubtful that model will attribute*
35 *so drastically different scaling factors for this shafts (for II the estimated emission is 0 while for IV and IX it is close 20 kt/year) if it works correctly. I, personally measured emission by this shaft in the year 2018 at the comparable level with other two, however, Gaussian plume model is also unreliable for such estimations – so I can't reveal it for comparison.*

*Piast IV (5kt/y - unc. 100%)– this is strictly non-methane coal mine shaft, where the amount of VAM is negligible like an-*
40 *other mine in the region (Ziemowit). For this reason, the result of scaling factor optimization is unreliable for both shafts (Piast IV and Ziemowit I (15kt/y - incorrectly). Jaworzno-Traugutt (30kt/y – incorrectly) – Currently the shaft should be named Sobieski-Traugutt as the mine is under the control of Tauron Wydobycie company and was renamed to ZG Sobieski. The coalbeds excavated in this mine are methaneless, it is non-methane mine. The value proposed by the authors is exaggerated by few orders of magnitude as it would require a high content of methane in the coal, which is absolutely not a*

*case here [e.g.: chapter 1.1.6.5, http://bip.katowice.rdos.gov.pl/files/obwieszczenia/53136/Obwieszczenie_RDOS_Katowice_WOOS_4235_5_2015_KC_21. Murcki Czułów (40kt/y – incorrectly) – Coal mine Murcki was connected with coal mine Staszic and subsequently closed. It is not operative now but its shafts are used for ventilation of part of the Staszic excavation works. The attribution of methane release to basically closed mine might, in this case, be reasonable. However, decision of the SRK (current owner of Murcki) and Staszic mine – the Czułów shaft has been closed in 2015 and since 2016 hasn't been used for ventilation purpose anymore (shaft was filled with gravel and clay) and finally in August 2018 it was dismounted [https://nettg.pl/news/152132/gornictwo-runela-wieza-szybu-czulow]. It is not possible that this amount of methane would be released by this shaft in June. So, the model has optimized the scaling factor wrongly, and surprisingly this was the highest methane emission rate attributed to any of the shafts. Murcki Zygmunt (0kt/y – incorrectly)– In opposition to shaft Czułów, this shaft is still operating and took all the ventilation work in relation to new longwalls work by Staszic mine in one of the deepest coalbeds containing a large amount of methane. In this case, however, the model has attributed scaling factor to zero with no methane emission. Comments regarding few westerly located shafts: Pniówek IV (0kt/y – incorrectly)– The active shaft belonging to one of the biggest methane emitting mines over USCB. Direct measurements recorded in the same time when the flight was performed allow to confirm that this shaft was operating and releasing VAM (methane associated with ventilation) on usual level (approx. 0.2% CH4 with 10000m3/min of air). For an unknown reason, the scaling factor reduced this emission to negligible values. Chwalowice V (17kt/y – might be correct) – The shaft belongs to the mine with two levels of exploitation. Shaft V was renovated in the year 2009 due to the demands of a new coalbed being planned for excavation. For this reason the shaft release much more methane than shaft no.VII which ventilates the older coalbeds containing less methane. Here, the proximity of the shafts to flight route allowed for the rational application of optimized scaling factors. However, methane emission of the mine is at the lower or same level as Marcel, Rydultowy mines, where deeper coalbeds are excavated (contain more methane). Summarizing, in the eastern part of USCB there is a mismatch of emission attributed to particular shafts at the summary level of 100Kt/y with the biggest value 40kt/y for the single shaft. I agree with authors that the momentary value should not be directly compared with yearly release reported by the coal mines as the day to day (or even hour to hour) emission my change by 100% (e.g. increase from 10 to 20 m3/min). For this reason, all the modelled values should be expressed in short time units (minutes or hours). But attributing large emission to a closed shaft or non-methane shafts are beyond of such explanation. In my opinion, the authors overestimate the accuracy of the dispersion model by an order of magnitude. Obviously, it has not enough spatial resolution to distinguish between particular shaft in case of single-wall pattern measurement.*

This is very valuable information and it is nice to see, that even under the very challenging situation of only 10° mean wind direction delta between the morning and afternoon flights, the algorithm is able to determine a magnitude of emissions from the shafts to match your information. We would like to encourage you to share your findings or your measurements with the scientific community as these data may be of interest for many others not reaching this comment. Based on your advice and the suggestion of Reviewer #1 we included this information into our a-prioris as "local information". The a-prioris for the closed mining shafts as indicated above have been set to $1 \, \text{kt} \, \text{yr}^{-1}$. The change of a-prioris, coupled with introducing correlations in the observational covariance (as suggested by Reviewer #1) led to some negative correlations in the a-posteriori correlation matrix. Furthermore uncertainties for individual shafts are enhanced compared to the version neglecting correlations. Lagrangian particle dispersion models like the one used in this study apply griding only after the simulation and only on the simulation output. Hence, the only spatial resolution that comes into play is the one of the diluted plume measured at the aircraft and the gridded meteorological input data. If you refer to the gridded input data, than achieving better simulation results would require knowledge on the turbulence in one grid cell, which is not available. In the end further campaigns and further research is needed to increase the level of confidence on the model side and to gather data at different wind conditions.

*I would like to give the appreciation for the authors to organize the COMET campaign and invest a lot of time to obtain great and novelty results which certainly are valuable material for publication.*

---

## Author Comment (AC3) · 17 Mar 2021

**In response to Anonymous Referee #1 comments from February the 5th, 2021.**

*The manuscript by Kostinek reports an interesting case study of top-down greenhouse gas emission quantification on the region scale using airborne observations, sophisticated atmospheric transport simulations and an inverse modelling framework. They estimate the CH4 emissions from coal mining shafts in Upper Silesia, one of the CH4 emission hot spots in Europe. The*
5  *research topic is very relevant regarding greenhouse gas emission mitigation strategies under the Paris Agreement and as such deserves publication. The study design, the applied transport simulations and partly the inverse modelling approach are sound and mostly presented well in the manuscript. The authors took great care to assure adequate atmospheric transport simulations and made an important effort to characterise the connected uncertainties, adding some valuable new concepts to the field. There are two major concerns, 1) regarding the feasibility of assigning emissions to individual ventilation shafts given*
10  *the flight observations taken at considerable distance from individual sources and 2) concerning the omission of CH4 sources other than coal mines. These issues are detailed below and an answer will probably require additional analysis and revisions of the manuscript. However, I would encourage the authors to address these additional points, so that their very valuable analysis can be published in ACP.*

Dear Referee,
15  We want to thank you very much for this excellent review and the detailed, helpful comments on the analysis presented in the manuscript. We greatly appreciate your work involved with this review. The comments include very valuable concepts, that required additional analysis but ultimately improved the manuscript significantly. We hope to have answered your comments to your fullest expectation.

***Major comments***
20  *Inverse modelling method: I have two major concerns with the emission estimation method.*

*1) Emission attribution to individual shafts The first concern is the attribution of emissions to individual facilities and shafts. Looking at figure 7 and 8 but also at the names of the shafts in figure 10, it is clear that many of the individual shafts cluster around individual mines at distances not much more than a kilometer. Furthermore, a lot of the locations actually line up with the main wind direction. In this situation it seems to be virtually impossible to estimate emissions from individual shafts from*
25  *the presented observational data which was only taken at a single downwind curtain. The results presented in figure 10 are most likely a fine example of overfitting the observations and obtaining a 'noisy' a posteriori result. The problem is also apparent from figure 11, which seems to indicate that although there is some sensitivity to all emission shafts, sensitivity is much larger for certain shafts than for others. Given the observational data, the problem cannot be overcome in a general way, but at least the covariances in the inversion should be designed in a way that will limit overfitting and the overinterpretation of emission*
30  *results. I would suggest that the authors modify the design of their a priori covariance. Currently, they only include diagonal elements. Hence, they assume uncorrelated uncertainties even for shafts from the same facility. I think it would be useful to explore by how much the results change if correlated a priori uncertainties for shafts from the same facility and/or shafts a shorter distances would be introduced (positive off-diagonal values in the a priori covariance matrix). Furthermore, the a posteriori covariance matrix should be explored in order to see if many of the emitters actually show negative covariances to*
35  *one another. This would indicate that there remained a large uncertainty as to which shaft the emissions had to be assigned. In general, the manuscript should better highlight that the uncertainty on the shaft-level remained large and that the observational data is to limited for a more specific estimate. In this context, it would also be good to present the shaft level emissions on a map and compare with spatial inventories. This issue was also raised in the comment by J. Necki and he provides valuable discussion on a per shaft basis that should not be ignored in a revision of the manuscript.*

40    Based on your advice we included "local information" as indicated in the short comment by Jaroslaw Necki into our a-prioris, by setting the a-prioris for the closed mining shafts to a very small number ($1\,\mathrm{kt\,yr^{-1}}$). In general we agree that our

Sect. 4.5 might have mislead the reader to think that we can accurately estimate individual shafts from two downwind walls. These estimates are certainly associated with significant uncertainty and correlations (as indicated by the inversion diagnostics), yet we are confident, that it is useful to include all available a-priori information into our state vector. With mission planning further optimized for the Bayesian inversion from airborne in situ data, as presented in this manuscript, these uncertainties can potentially be narrowed down in future campaigns. We rephrased Sect. 4.5 accordingly. In general, subregional emission estimates require at least two flights associated with different wind conditions. During single flights several emission sources are masked by sources closer to the point of measurement. In the present scenario, the large number of sources enables regional estimates for single flights, albeit subregional estimates will vary significantly and are not reliable for the reason mentioned above. For this reason we use two flights (morning and afternoon) that are close in time to reduce the impact of daily changes in emission rates yet allow for the determination of subregional emission estimates, albeit under the very challenging situation of only $10°$ mean wind direction delta between the morning and afternoon flights.

It is certainly true, that several shafts cluster around individual mines at distances not much more than a kilometer. However, that does not directly yield the amount of correlation of emissions from individual shafts. To our knowledge, methane emission strongly depends on the underground location where actual excavation is taking place. Furthermore individual shafts may also be used for fresh air supply. Therefore it is doubtful if the conceivable additional constraints (a-priori correlation) are appropriate or not. Yet we think it is a good idea to explore by how much the results change if correlated a-priori uncertainties for shafts from the same facility are introduced. The result for a $+.5$ correlation of the mines corresponding to clusters is shown in Fig. 1. As expected, the correlated a-priori covariance leads to a smoothing between individual shafts. The total emission

[Figure]

**Figure 1.** Left: +.5 correlated a-priori covariance. Individual clusters can be identified from the off-diagonal elements. Right: Impact of the correlated a-priori on emission estimates from individual sources.

rate changes by less than $< 0.5\%$, albeit there is some change in individual emission rates. In general the introduced additional a-priori information suggested by a-priori correlations remain to be verified, but we decided to follow this referee advice.

In contrast to the a-priori covariance matrix, correlations in the observational covariance matrix have a significant impact on individual estimates. The observational covariance had been estimated as a diagonal matrix in the first version of this manuscript neglecting correlations. In order to compensate for tempo-spatial-autocorrelation we simulated a puff release to check the impulse response of the simulation on a one-second release from individual sources. We sampled simulated observations from this puff release at the aircrafts location in space and time and computed the Autocorrelation function (ACF) from a single dispersed puff. The exponential decay of the correlogram suggested to augment the observational covariance matrix with a first order autoregressive model AR(1) structure with $\varphi = 0.7$. Fig. 2 shows a zoom on the first $400 \times 400$ elements of this $14936 \times 14936$ matrix to highlight the introduced off-diagonal elements. The modified observational covariance matrix led to a decrease in degrees of freedom for signal from $d_s = 48$ to $d_s = 32$, indicating that now more information stems from the a-priori. The right panel of Fig. 2) shows the a-posteriori correlation matrix as deduced from the covariance matrix. From the

[Figure]

**Figure 2.** Left: Observational covariance augmented with a first order autoregressive model AR(1) structure with $\varphi = 0.7$ Right: A-posteriori correlation matrix deduced from the a-posteriori covariance matrix.

a-posteriori correlations matrix, there is some negative correlation as expected by the reviewer. In general, the uncertainties for individual shafts are enhanced as can be seen from the updated Fig. 3. This figure is to be compared to Fig. 10 in the original version of this manuscript.

[Figure]

**Figure 3.** Emission estimates $\Phi_i$ (blue bars) in kt yr$^{-1}$ for 74 individual mining shafts using the morning and afternoon flight of June 6th, 2018. Slim green bars are the reported yearly average values for each mining company (E-PRTR 2017) evenly distributed among the respective ventilation shafts. The orange error bars stem from the quadrature sum of the statistical uncertainties $\epsilon_i$ (computed from the parameter covariance matrix $\hat{\mathbf{S}}$) and the uncertainties $\sigma_{ensemble}$ derived from a variational ensemble with systematically perturbed parameters.

*2) Neglect of other than coal mine emissions In section 2 it is discussed that EDGAR assigns about 14 % of total CH4 emissions in the area to non-coal mine emissions. The authors largely ignore these emissions in their analysis. Only in section 4.4 the possibility of emissions from the city of Krakow are discussed but ultimately these were also not included for the inverse modelling. Although 14 % may not seem a very large fraction and one could argue that any kind of mis-attribution is covered*

5  *by the uncertainty estimates, it will still strongly depend on the distribution of these missing emissions relative to the sampling locations. Their impact on the concentration observations may well have been much larger if sources were closer to the flight track than the coal mine emissions. Since the authors also find that total emissions for the region are considerably smaller than what was reported in EDGAR, the fraction of non-coal may also be larger than in EDGAR. Seasonality of emissions may also play a role here. Since this is early summer, temperatures were above annual average, possibly leading to larger than annual*

10  *average emissions from microbial sources like landfills, waste water treatment, manure management, etc. There seems to be a large discrepancy with EDGAR in terms of shafts and emission locations as well (red grid cell in the lower left corner of the encircled area close to Ostrava). Are these actually coal mine emissions in EDGAR? Looking at the area, one can also see a larger reservoir north-west of Bielsko Biala. Could natural emissions from this reservoir also play a role in the poorer agreement at the eastern end of the curtain flight? I strongly suggest that the authors reconsider their neglect of the noncoal*

15  *mine emissions. They should obtain inventory data of these emissions, possibly not just from EDGAR (see comment below on inventories) but other more resolved inventories or from local information (see comment by J. Necki). With these emissions another FLEXPART forward run should be conducted and the resulting concentration either be removed from the observation vector before the inversion or an additional scaling factor for non-coal mine emissions should be introduced in the inversion.*

The neglect of non-fuel-exploitation can arguably lead to the fuel-exploitation emission estimate to be biased towards higher

20  values. Thus, additional partitioning into non-fuel and fuel-exploitation emissions does only further increase the projected discrepancy. As suggested, we re-analyzed our data, this time including non-fuel-exploitation sources. To this end, we added a figure to a revised version of this manuscript on methane emissions grouped by sector as provided by EDGAR to depict the spatial distribution of these sources and to showcase their small magnitude versus coal-mine emissions. The EDGAR v4.3.2 inventory includes information on sectorial partitioning of $CH_4$ emissions with non-fuel-exploitation making up for approxi-

25  mately $\sim$14 % of total annual $CH_4$ emissions in the USCB. From these $\sim$14 % approximately 90 % are attributed to the five sectors: Solid waste landfills, Energy for buildings, Waste water handling, Enteric fermentation and Oil refineries and transformation energy. Most emitters are weak in comparison to fuel-exploitation and uniformly distributed. Uniformly distributed emitters will be canceled out by the background subtraction and will hence not influence the emission estimate. Some stronger non-fuel-exploitation sources are however apparent. We added these source tiles (threshold $\geq 4\,\mathrm{kt\,yr^{-1}}$) to our FLEXPART-

30  WRF simulation. This led to a partitioning of the emission estimates into non-fuel and fuel exploitation emission estimates for the USCB. The included non-fuel-exploitation sources are estimated with $27\,\mathrm{kt\,yr^{-1}}$ and $31\,\mathrm{kt\,yr^{-1}}$ for the morning and afternoon flights, respectively. This corresponds with the EDGAR a-priori ($33\,\mathrm{kt\,yr^{-1}}$) for non-fuel-exploitation to within 20 %. The derived fuel-exploitation emission estimates amount to $451\,\mathrm{kt\,yr^{-1}}$ and $423\,\mathrm{kt\,yr^{-1}}$ for the morning and afternoon flights, respectively. The small deviations of less than 5 % from the original run suggest a robust estimate for the USCB region.

35  Nevertheless, including non-fuel-exploitation did make up for a substantial improvement, as this partitioning is only possible with sophisticated atmospheric simulations. This could lead to large errors in regions where non-fuel-exploitation is more relevant than in the USCB and/or for emission estimation approaches not involving atmospheric models. This lesson-learned will definitely be propagated to upcoming studies.

**Minor comments**

40  1. **p4: Regarding the use of the EDGAR inventory I would like to question if this is really the best available bottom-up inventory for the area. First of all, there is newer version of EDGAR available (v5.0 GHG), which explicitly lists CH4 from coal exploitation as a separate category and is available for a more recent year (2015) than EDGAR 4.3.2. Furthermore and as part of the EU project CHE, TNO has compiled higher resolution (6 km x 6 km) inventories for Europe. They may be better suited than EDGAR (see https://www.che-project.eu/sites/default/files/**

**2019-01/CHED2-3-V1-0.pdf; data usually available on request). This is not only important for the final comparison of obtained emission estimates but also relates to the question if and how non-coal emission need to be treated in the inversion framework.**

We have recently published a comparison between several available inventories for the USCB in Fiehn et al. (2020b) including inventories like E-PRTR, Scarpelli $CH_4$, CAMS-REG v3.1, EDGAR v5 and GESAPU. They all have their intrinsic advantages and disadvantages. Instead of re-iterating over the available inventories, we decided to showcase the large discrepancy between two well-known and well-established inventories to highlight the necessity of improving on bottom-up derived emission inventories via top-down GHG emission quantification.

2. **p5, l109: Here it is mentioned that an upwind concentration is subtracted from the downwind measurements. Later on a different method for background subtraction is described. What was really used?**

Some words were missing in this sentence, misleading towards the assumption, that upwind leg mixing ratios were subtracted from the downwind mixing ratios. We added the missing words: "[...] showing a fairly homogeneous CH4 inflow into the area of interest, thus allowing for subtracting an out-of-plume background (as described in Sect. 4.4) from the measured mole fractions downwind of the mines. [...]"

3. **p5, l113: Why is detrainment/entrainment important to this study? The FLEXPARTWRF simulations don't exclude detrainment/entrainment processes or PBL growth. Detrainment/entrainment would be more of an issue for a mass balance approach.**

This sentence is a historic residue, as the study was first based on a mass balance approach and only afterwards enhanced with high resolution particle dispersion simulations. We removed the obsolete sentence in a revised version of this manuscript.

4. **p5, l117: While an estimate of the morning PBL height is given, its height is not mentioned for the afternoon flight. Please add. Maybe also comment on the growth of the PBL height between the two flights and how this relates to the question of detrainment/entrainment.**

We added the missing information on the PBL height for the afternoon flight: "[...] During this flight, we observed an latitudinally inclined PBL with an approximate depth of $1.7\,\mathrm{km}\,\mathrm{a.M.S.L}$ in the northern section and $1.3\,\mathrm{km}\,\mathrm{a.M.S.L}$ towards the south. [...]"

5. **p7, L148f: Were the Doppler soundings the only observations that were nudged? What about other standard synoptic observations in the area?**

Yes, the Doppler soundings were the only observations fed into the observational four dimensional data assimilation. According to previous studies (see e.g. Cambaliza et al. (2014)) emission estimates obtained from airborne in situ data, are primarily affected by errors in wind speed. This is also apparent from our previous uncertainty analysis published in Fiehn et al. (2020a), where uncertainty in wind speed makes up for a large fraction of total uncertainty. For this reason, and because the used meteorological input data *NCEP GDAS/FNL 0.25 Degree Global Tropospheric Analyses and Forecast Grids* (GDAS/FNL, 2015) already assimilates global observational data, we refrained from nudging with conceivably the same observations again.

6. **p8, l158ff: Does this mean that 3Dvar and nudging were applied to the same observational data? That would not make sense in my view as the same information gets used twice. Rather use 3Dvar with smaller error covariance if the pull of the observations seemed too weak and such smaller uncertainties could be justified. Also, how were observational error covariances determined exactly?**

3DVar only assimilates observations temporally close to the 3-hourly available NCEP GFS analyses. As such 3DVar provides the initial conditions for the next 3 hours of WRF simulation. Meanwhile OBS-FDDA assimilates the data continuously during the WRF run as detailed in the manuscript. Smaller error covariances would not help. Instead, if analyses were available at smaller time intervals, the increased number of 3DVar runs could positively affect the results.

7. **p8, l167: Are these numbers the root mean square errors between model and observations for the 1-Hz sampling?**

These numbers correspond to the standard deviation of the residuals between simulated and 1-Hz sampled observational

data. We clarified this in a revised version: "[...] Simulated data, extracted at the aircraft positions in space and time, agree with 1 Hz observations of wind speed and direction to within an RMSE of $\pm\, 0.7\,\mathrm{ms}^{-1}$ $(1\,\sigma)$ and $\pm\, 5\,^\circ$ $(1\,\sigma)$, respectively. [...]"

8. **p8, l171f: How can this apparent offset in pressure be explained? Difficult to believe that the models (WRF nested in GFS) are off by that much, especially since the wind seems to match very well. Was there any comparison to other surface pressure data? Concerning the temperature offset: Does this vanish when you calculate potential temperatures? And same question as for pressure: were there any ground based measurements to compare to?**
We do not yet understand the reason for these offsets. The temperature bias does not vanish for potential temperatures. As described above, the Doppler soundings were the only observations used for data assimilation. We did not check with ground based measurements as these should be already assimilated in the meteorological input data *NCEP GDAS/FNL 0.25 Degree Global Tropospheric Analyses and Forecast Grids* (GDAS/FNL, 2015).

9. **p9, inversion method: I got confused by the description here. First, a non-regularized least square equation is presented for flux optimisation (eq. 1). Then regularization using a priori information and Bayes' theorem is advocated. To my understanding the resulting equations 4 and 5 only require a simple matrix inversion for solving for the a posterioir state. However, from line 211 onwards the application of a non-negative least square solver is presented. The latter is probably applied to equation 1, yielding a positive solution for x. However, if this was the case, I don't see why further analytical solutions to the cost function are presented in 4 and 5. I assume I am missing an important point here and would like the authors to clarify. If only a positively constrained solution for equation 1 is obtained I would think the results are even more overfitted as already mentioned above, since no additional a priori constraint on the individual sources would have been used. The description in the results section strongly suggests that this was the case. In equation, 4 I also think the last term $Kx$ should also be $Kx_a$ instead (see Tarantola eq. 3.37 or Jacob eq. 23).**
The introduction with a non-regularized least squares approach is intended to introduce and clarify the method. In the present case however, it is not necessary for the subsequent steps and has therefore been removed in a revised version of this manuscript. If the matrix $\mathbf{KS_aK^T} + \mathbf{S}_\epsilon$ is invertible, then a matrix inversion does the trick, albeit it finds a solution that also includes negative sources. These negative sources correspond to significant $CH_4$ sinks, and hence are treated as unphysical. The NNLS algorithm is used to discriminate sinks from the solution vector $\mathbf{x}$, yet it is not applied to Eq. 1 but to the MAP cost function Eq. 3 and therefore includes a-priori information on the emitters.
We revised the relevant parts of the manuscript to: "[...] Following a maximum a posteriori (MAP) approach, the scaling coefficients $x_i$ can be found for each of the $n$ modeled sources $\varphi_i$ and for each of the $m$ observed enhancements $y_j$ making use of a-priori information $\mathbf{x_a}$ on the emissions of the individual shafts. Following Bayes' theorem the MAP solution is given by the minimum of the cost function (Tarantola (2004); Jacob (2007); Rodgers (2000))

$$
\begin{aligned}
J\left(\mathbf{x}\right) &= \left(\mathbf{x} - \mathbf{x_a}\right)^T \mathbf{S_a}^{-1}\left(\mathbf{x} - \mathbf{x_a}\right) \\
&+ \left(\mathbf{y} - \mathbf{Kx}\right)^T \mathbf{S}_\epsilon^{-1}\left(\mathbf{y} - \mathbf{Kx}\right)
\end{aligned}
\tag{1}
$$

with later defined a-priori and observational error covariance matrices $\mathbf{S_a}$ and $\mathbf{S}_\epsilon$, respectively. The MAP solution can be found by solving for $\nabla_x J\left(\mathbf{x}\right) = 0$ and is given by

$$
\hat{\mathbf{x}} = \mathbf{x_a} + \mathbf{G}\left(\mathbf{y} - \mathbf{Kx}\right)
\tag{2}
$$

with the gain matrix

$$
\mathbf{G} = \mathbf{S_aK^T}\left(\mathbf{KS_aK^T} + \mathbf{S}_\epsilon\right)^{-1}
\tag{3}
$$

By exploiting the averaging kernel $\mathbf{A} = \mathbf{GK}$ the number of degrees of freedom for signal $d_s$ can be computed as

$$
d_s = tr\left(\mathbf{A}\right)
\tag{4}
$$

This number describes the reduction in the normalized error on $\mathbf{x}$ introduced by the available observations and hence provides a measure for the improvement in knowledge of $\mathbf{x}$, relative to the a-priori, due to the observations.

The total emission estimate $\Phi$ in units $\mathrm{kg\,s^{-1}}$ follows from the scaled sum of the individual contributions $\varphi_i$

$$\Phi = \sum_{i=1}^{n} x_i\,\varphi_i \tag{5}$$

Here, the Non-Negative Least Squares (NNLS) algorithm (Lawson and Hanson, 1995) has been used to minimize the MAP cost function subject to the constraint $\mathbf{x} > 0$. This constraint is equivalent to the absence of negative sources. [...]"

10. **p16, l336: Here the total uncertainty of the emission estimate is presented as the sum of the 'systematic' uncertainty (which I assume results from the a posteriori covariance; eq. 8) and the spread obtained from the sensitivity simulations. Why are these uncertainty terms not added quadratically?**
As for any independent variables, the standard deviation of the sum is the quadrature sum of the individual standard deviations. We revised the respective text sections in the manuscript.

11. **Section 4.3: The way the covariance matrices for the a priori and the observation/model error are constructed most likely oversimplifies the true nature of the involved covariances and may lead to overfitted results. First, and already mentioned in the main comment above, the a priori covariance should acknowledge the fact that the a priori emission uncertainties will be correlated. This is true for shafts belonging to the same mining complex but may also be true for spatial distances between shafts. As mentioned above, I would suggest introducing off-diagonal elements in the covariance matrix to honor this fact. This would certainly lead to a smoothing out of the emissions across different shafts but is a more realistic approach. Furthermore, the observation/model covariance does not include off-diagonal elements either. However, the 1-Hz observations are certainly not independent from each other since they contain tempo-spatial autocorrelation. The latter will also be present in 1-Hz model residuals. I would suggest to explore this auto-correlation in the residuals and add a temporal correlation length to the observation/model matrix accordingly. Adding these off-diagonal elements will probably reduce the impact of the observations on the a posteriori results, reflecting that they are not really independent from each other. Another way to get rid of the autocorrelation would be temporal averaging of the observations before using them in the inversion. This has its merits as well as it would also bring the spatial resolution of the observations closer to those of the transport model.**
This comment is basically a detailed version of major comment #1. We replied above to major comment #1.

12. **p11, l231: How was the transport model uncertainty estimated concretely? As the standard deviation of simulated concentrations from the 8 ensemble members?**
The transport model uncertainty was estimated from 8 sensitivity runs, where the respective variables were perturbed in one or the other direction globally, but not perturbed by random noise with the given sigma width. Changes in the manuscript are described below at Comment #15.

13. **p12, l248: Why not use the measurements from the upstream flight segment as background? The comparison with the model output seems to indicate that the overall plume was wider than the flight segments.**
Due to the limited available flight time, the upwind leg is flown at a single altitude only. Subtracting the observations from this flight leg from the downwind wall pattern would require a Lagrangian simulation of the mixing ratioes during the upwind leg projected onto the downwind wall location. While possible, it would introduce significant additional uncertainty due to the single flight leg. For this reason we refrained from subtracting a Lagrangian propagated background and decided to use the out-of-plume background on both sides of the downwind wall.

14. **p14, l296: This sentence largely repeats the result from the previous sentence (EDGAR being much larger than the current estimate).**
The sentence is obsolete and has been removed.

15. **section 4.6: After reading the first few sentences, it was not clear to me how an uncertainty quantified by sigma was adopted in the transport model. I guess figure 11 makes it clear that 8 sensitivity runs were done where the respective variables were perturbed in one or the other direction globally, but not perturbed by random noise with the given sigma width. This should be made a bit clearer from the beginning.**

We revised the introductory sentences of this section to make the derivation of the systematic transport model uncertainty a bit clearer: "[...] The influence of several variables on the total flux estimate $\Phi$ has been computed from 8 sensitivity runs with symmetrically perturbed parameters. The systematic transport model uncertainty is subsequently estimated as the standard deviation of this ensemble. [...]"

16. **p15, l325: If I understand correctly, original horizontal wind speeds as output from WRF were increased/decreased by 0.9 m/s. In doing so, the local mass balance of the wind field may well be destroyed as vertical wind speeds were not adjusted (correct?). This may lead to errors in the transport description of the LPDM. Have you given this any thoughts? Probably the impact was not to large and since this is only presented as a sensitivity case it is of less importance, but it may have lead to larger discrepancies from the reference run than anticipated. A similar question for the PBL height. Is the latter taken from WRF or is the diagnostic calculation taken from FLEXPART? When increasing the PBL height just in FLEXPART vertical mixing in FLEXPART may then bring model particles to altitudes that in WRF are not part of the PBL and as such may have a distinctly different flow direction as flow in the PBL. As a consequence the differences to reference run may be larger than in a case where WRF PBL heights were larger/smaller. Hence, your change in the PBL height may give a slightly more pessimistic (larger) uncertainty.**

We are aware of, that by not adjusting the vertical wind fields, the local mass balance of the wind field may be jeopardized leading to larger residuals and hence larger uncertainties. However, as this is only a sensitivity analysis it is of minor importance here. In contrast, the PBL height has implications for all runs and hence also for the emission estimate. To retain a local mass balance of the wind field we do not use the diagnostic PBL height calculations from FLEXPARTWRF but feed the LPDM with the PBL height as simulated by WRF.

We added this info in a revised version of the manuscript: "[...] FLEXPART-WRF version 3.3.2 (Brioude et al., 2013) was used to model the exhaust plumes of known emitters forward in time using the meteorological data (including PBLH) obtained from the WRF simulations described above (see Sect. 4.1) as a driver. [...]"

17. **p16, l23: Here it is mentioned that the statistical uncertainty was estimated from eq. 7 and 8 and it is referred to elements $e_i$, which are the diagonal elements of the a posteriori covariance matrix. What about the off-diagonal elements of this matrix? Were the taken into account for the total uncertainty estimate?**

Off-diagonal elements are included for the regional emission estimate uncertainty. Including off-diagonal elements of the a-posteriori covariance matrix for single entries of the state vector would require a change into a basis where all off-diagonals vanish, if such a basis exists at all.

18. **Figure 12 and use of Jacobian: If I understand correctly, what is shown in figure 12 is the matrix K containing the elements $dy_j/dx_i$. However, the term Jacobian is also used in the manuscript for grad(J(x)). But J is not a vector-valued function and as such grad(J) is not a Jacobian. Please clarify.**

The Jacobian we are referring to throughout the text is the matrix $\mathbf{K}$. We removed the reference to grad(J) as the Jacobian in a revised version of the manuscript.

**Technical comments**

19. **p2, l45. Karion et al. is missing a publication date.**

The missing date has been included in a revised version of this manuscript.

20. **Figure 1: The line indicating the afternoon flight is more orange than red (as described in caption). It looks like the map is showing total EDGAR emissions. How does the distribution of non-coal mine emissions look like?**

We made the line in Fig. 1 look more reddish. The distribution of non-coal mine emissions has been included in a revised version of the manuscript.

21. **Figure 3: Please label the WRF domain in the figure according to their definition in the text.**
    WRF domains have been labeled according to their definition in the text in a revised version of this manuscript.

22. **Figure 4+5: Please use the same colors for the different WRF runs. The legend is fairly small in both cases and needs to be enlarged, possibly put to the right of the sub-panels.**
    Figure 4+5 have been adjusted accordingly in a revised version of this manuscript.